# The RNA quality control pathway nonsense-mediated mRNA decay targets cellular and viral RNAs to restrict KSHV

Yang Zhao [ID] [1], Xiang Ye[1], Myriam Shehata[1], William Dunker [ID] [1], Zhihang Xie[1] & John Karijolich [ID] [1,2,3,4,5 ✉]

Nonsense-mediated mRNA decay (NMD) is an evolutionarily conserved RNA decay mechanism that has emerged as a potent cell-intrinsic restriction mechanism of retroviruses and positive-strand RNA viruses. However, whether NMD is capable of restricting DNA viruses is not known. The DNA virus Kaposi's sarcoma-associated herpesvirus (KSHV) is the etiological agent of Kaposi's sarcoma and primary effusion lymphoma (PEL). Here, we demonstrate that NMD restricts KSHV lytic reactivation. Leveraging high-throughput transcriptomics we identify NMD targets transcriptome-wide in PEL cells and identify host and viral RNAs as substrates. Moreover, we identified an NMD-regulated link between activation of the unfolded protein response and transcriptional activation of the main KSHV transcription factor RTA, itself an NMD target. Collectively, our study describes an intricate relationship between cellular targets of an RNA quality control pathway and KSHV lytic gene expression, and demonstrates that NMD can function as a cell intrinsic restriction mechanism acting upon DNA viruses.

[1] Department of Pathology, Microbiology, and Immunology, Vanderbilt University Medical Center, Nashville, TN 37232-2363, USA. [2] Department of Biochemistry, Vanderbilt University School of Medicine, Nashville, TN 37232-2363, USA. [3] Vanderbilt-Ingram Cancer Center, Nashville, TN 37232-2363, USA. [4] Vanderbilt Institute for Infection, Immunology and Inflammation, Nashville, TN 37232-2363, USA. [5] Vanderbilt Center for Immunobiology, Nashville, TN 37232-2363, USA. ✉email: john.karijolich@vanderbilt.edu

Viruses rely on cellular machinery for various aspects of gene expression, including transcription, RNA processing, and translation of their RNAs. Thus, viral gene expression is regulated by many of the same quality-control (QC) processes as the host. However, because of the highly compact nature of viral genomes and the encoding of a large number of viral RNAs within a relatively small amount of genomic space, gene expression strategies that ignore host QC processes are sometimes employed. Consequently, in addition to functioning in the QC of gene expression select mechanisms have emerged as having powerful antiviral roles[1].

Nonsense-mediated mRNA decay (NMD) is an evolutionarily conserved RNA QC mechanism and is present in all eukaryotes examined to date[2]. NMD is translation-dependent and is triggered by ribosomes terminating at a position deemed to be aberrant by the NMD machinery[3]. The best-characterized substrates of NMD are mRNAs that harbor normal or premature termination codons (PTCs) located 50–55 nucleotides (nt) upstream of an exon–exon junction[4]. However, additional substrates such as mRNAs that contain upstream open read frames (uORFs) and long 3′-untranslated regions (UTRs) can also be targeted[5–8]. Thus, NMD regulates steady state levels of both aberrant and wild-type transcripts.

Exon–exon junction complexes (EJCs), which is a complex of proteins deposited at the exon–exon junction in a pre-mRNA splicing-dependent manner, play a critical role in NMD substrate identification[9,10]. During translation elongating ribosomes remove EJCs that are located within ORFs[11]. RNAs upon which translation terminates upstream of an EJC are considered as aberrant and the RNA nucleates the formation of an NMD-activating protein complex[9,10,12,13]. The complex includes a set of regulatory factors called up-frameshift (UPF) 1–3 and additional factors in metazoans[12,14–16]. UPF1 is the key NMD-activating protein and its phosphorylation by the protein kinase SMG1 results in translational repression and commits the RNA to degradation[17–19].

While best characterized in RNA QC, NMD has emerged as a contributor to antiviral immunity in plants as well as against select human RNA viruses and retroviruses[20–27]. In particular, positive-strand RNA viruses appear to be most susceptible to NMD-mediated restriction[20–24,26]. However, whether NMD functions as an antiviral defense mechanism against DNA viruses is not known.

Here, we demonstrate that NMD contributes to restriction of the DNA virus Kaposi sarcoma-associated herpesvirus (KSHV). Lytic reactivation of KSHV, which is the etiological agent of Kaposi's sarcoma and primary effusion lymphoma (PEL), is significantly enhanced upon knockdown of NMD factors. Coupling RNA-sequencing and transcriptome-wide identification of phospho-UPF1 (p-UPF1) bound RNAs we define NMD substrates in the PEL cell line TREx-BCBL1-RTA. Moreover, we demonstrate that NMD targets both cellular and viral transcripts, including the mRNA encoding the viral transcription factor, replication, and transcription activator (RTA), to restrict KSHV reactivation. Unexpectedly, we identified an NMD-regulated pathway that couples activation of the unfolded protein response (UPR) with transcriptional activation of RTA. Collectively, our study describes an intricate relationship between cellular targets of an RNA QC pathway and KSHV lytic gene expression, and demonstrates that NMD can function as a cell-intrinsic restriction mechanism acting upon DNA viruses.

## Results

### Widespread pre-mRNA splicing within the KSHV transcriptome.
We recently reported a transcriptome-wide single-nucleotide resolution map of KSHV transcription start sites (TSSs) using RNA annotation and mapping of promoters for analysis of gene expression (RAMPAGE)[28]. In our analyses, we unexpectedly observed several unannotated introns, including many positioned downstream of a termination codon, rendering the transcripts potential NMD substrates. Emboldened by these observations we sought to comprehensively identify pre-mRNA splicing events in the KSHV transcriptome in two well established models of KSHV infection, namely, iSLK.219 and TREx-BCBL1-RTA cells. iSLK.219 cells are of clear cell renal cell carcinoma origin and are infected with a recombinant virus, KSHV.219, in which GFP, expressed from the elongation factor 1-α promoter, and RFP, expressed from the viral lytic gene PAN promoter, were integrated into the viral genome[29]. In contrast, TREx-BCBL1-RTA cells are an engineered cell line derived from an HIV+ patient with PEL[30]. A key feature of both models is the integration of a doxycycline (Dox)-inducible version of the major viral transcription activator RTA, and thus upon introduction of Dox into the cell culture media KSHV enters the lytic cycle.

To identify novel introns we performed ultra-deep high-throughput RNA-sequencing of RNA isolated from iSLK.219 and TREx-BCBL1-RTA cells in a latent state, or at 24, 48, 72, and 96 h, post-Dox-induced reactivation. We generated ~80 and ~150 million uniquely mapped reads for each time point in iSLK.219 and TREx-BCBL1-RTA cells, respectively. Leveraging the splicing aware algorithm, STAR, we identified many previously undescribed pre-mRNA splicing events. In fact, while only 27 splice junctions were previously reported, we observed 372 and 68 splice junctions in iSLK.219 and TREx-BCBL1-RTA cells, respectively (Fig. 1a and Supplementary Data 1 and 2)[31]. With the exception of one splice junction spanning the ORF41-50 locus, we identified all previously reported junctions[31]. To validate our analyses, we selected two novel introns, located within ORF46 and K10, and verified the splice junctions by Sanger sequencing of cDNA amplicons (Fig. 1b, c and Supplementary Fig. 1a). Sanger sequencing confirmed our bioinformatic analyses. Interestingly, many of the identified introns are positioned downstream of termination codons within polycistronic transcripts (Fig. 1d, Supplementary Fig. 1b, Supplementary Data 1 and 2), a situation that would render viral RNAs potential NMD substrates. For example, the novel ORF46 intron is located in the 3′UTR of the ORF48 transcript.

Pre-mRNA splicing is facilitated in part by splice sites that flank intronic sequences. Analyses of eukaryotic pre-mRNA splicing have revealed the 5′ and 3′ splice sites predominantly consist of GU and AG dinucleotide sequences[32,33]. However, functional non-consensus sequence splice sites are known to exist[34]. To determine whether KSHV splice sites harbor consensus 5′- and 3′-consensus nucleotides we performed kpLogo analysis on all identified splice sites. Indeed, kpLogo identified a prominent 5′- GU dinucleotide consensus sequence, as well as a preference for a G nucleotide 3′ of the splice-acceptor (Fig. 1e, f). The identification of a consensus 5′ splice-site as well as sequences that can serve as 3′ splice acceptors flanking KSHV introns suggests these events are facilitated by the cellular spliceosome.

EJCs are deposited at the exon–exon junction, thus we determined whether the EJC component eukaryotic translation initiation factor 4A3 (eIF4A3) is associated with spliced KSHV RNAs. To test EJC association with KSHV transcripts we performed eIF4A3 formaldehyde crosslinking RNA immunoprecipitation (fRIP) coupled to reverse transcription quantitative PCR (RT-qPCR) on lytic iSLK.219 and TREx-BCBL-RTA cells (Fig. 1g, h). eIF4A3 efficiently immunoprecipitated several KSHV RNA, including ORF50, K8.1, and ORF57, indicating the presence of an EJC on viral RNAs. In contrast, eIF4A3 fRIP did not enrich the small noncoding RNA vault RNA 1-1 (vtRNA1-1).

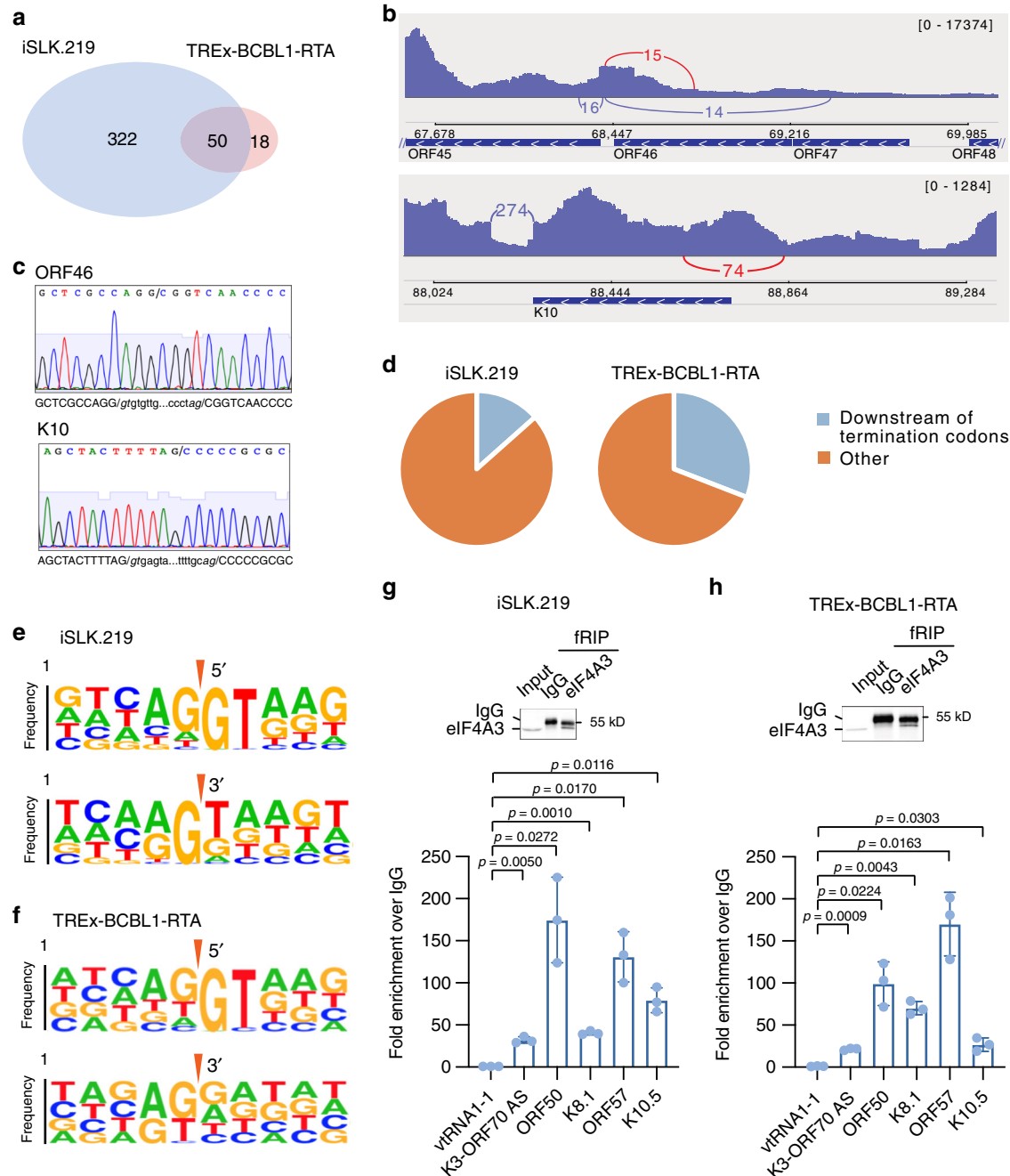

**Fig. 1 Identification of pre-mRNA splicing events in the KSHV transcriptome. a** Venn diagram illustrating the overlap of viral splicing events in iSLK.219 and TREx-BCBL1-RTA cells. **b** IGV view of pre-mRNA splicing events identified in ORF46 (upper panel) and K10 (lower panel). The novel pre-mRNA splicing events verified in **c** are labeled in red. **c** Verification of novel pre-mRNA splicing events in ORF46 and K10 by Sanger sequencing. **d** Pie chart depiction of the percentage of identified splicing events downstream of termination codons in viral transcripts. **e, f** kpLogo analysis of 5′ and 3′ splice sites in iSLK.219 (**e**) and TREx-BCBL1-RTA cells (**f**). **g, h** Verification of the association of viral transcripts with eIF4A3 by fRIP-qPCR in iSLK.219 (**g**) and TREx-BCBL1-RTA cells (**h**) 48 h post-Dox treatment. Upper panels are western blot of cell lysis (Input) and fRIP products (representative of three independent experiments). Lower panels are RT-qPCR analyses of eIF4A3 fRIP. Data are presented as mean values ± SD ($n = 3$ biologically independent samples). $p$-values were determined by the two-tailed Student's $t$-test. Source data are provided in Source Data file.

Collectively, these analyses have determined that the KSHV transcriptome is subjected to extensive pre-mRNA splicing and that the EJC is present on many viral RNAs.

**Knockdown of UPF1 facilitates KSHV lytic reactivation.** Given that KSHV expresses RNAs that harbor NMD-eliciting features, such as long 3′UTRs as well as 3′UTR introns, we hypothesized that NMD impacts KSHV lytic reactivation. To test this, we

determined the effect of UPF1 knockdown on KSHV spontaneous and Dox-induced lytic reactivation in iSLK.219 cells (Fig. 2a). Interestingly, even without Dox treatment we observed a minor increase of RFP-positive cells in UPF1-depleted cells (Supplementary Fig. 2a–d), and RT-qPCR analysis revealed an increase in viral lytic gene expression (Supplementary Fig. 2e). Moreover, consistent with the increase in viral gene expression we observed ~8-fold increase of KSHV virions in the supernatant of

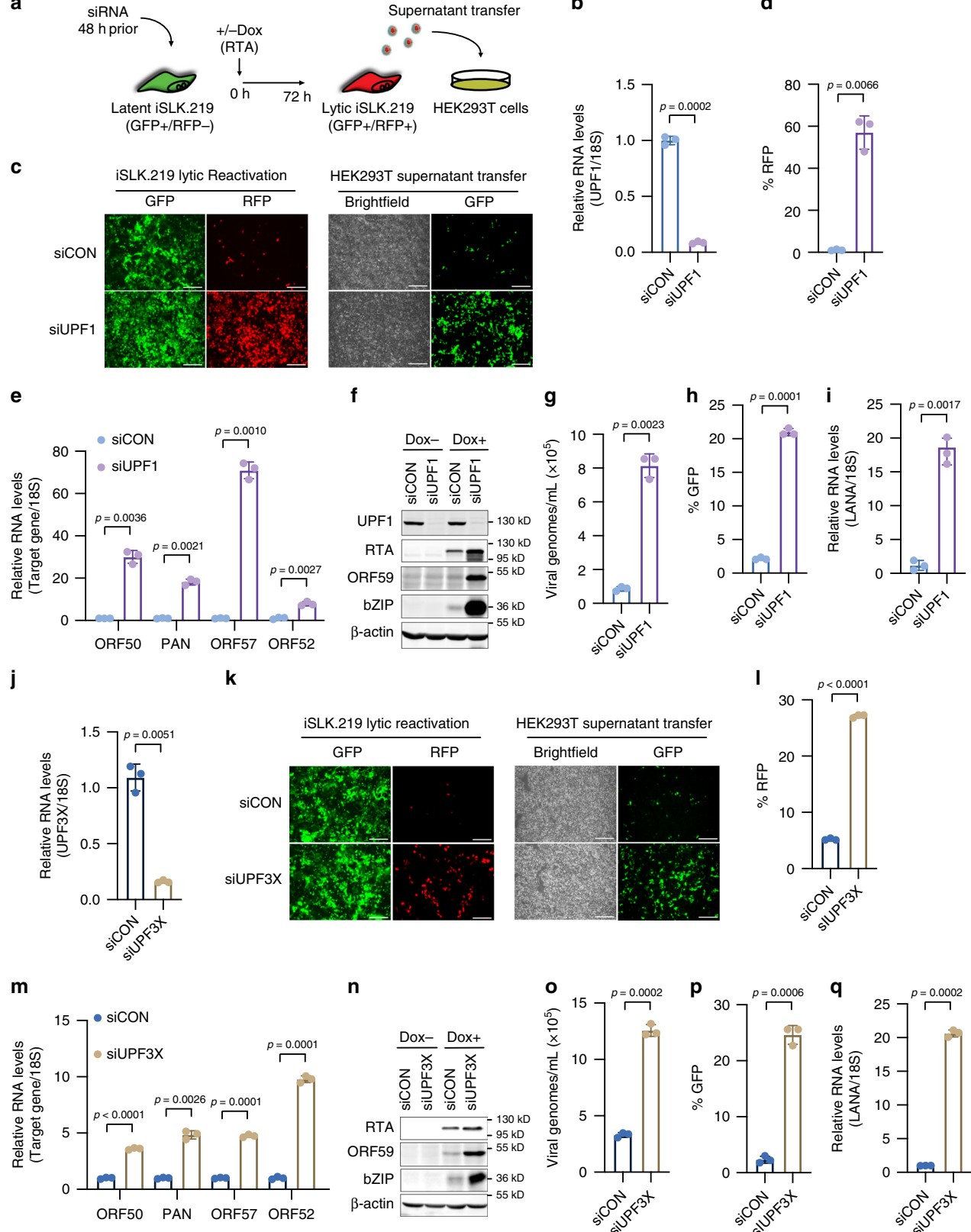

UPF1-siRNA (siUPF1) treated cells relative to control-siRNA (siCON) treated cells (Supplementary Fig. 2f).

While the increase in spontaneous reactivation was minor, Dox-induced lytic reactivation was robustly enhanced in siUPF1-treated cells relative to siCON-treated cells (Fig. 2b, c).

Flow-cytometry analyses of lytic cells (RFP-positive) demonstrated ~10-fold increase in RFP-positive cells 48 h post-Dox induction (Fig. 2d and Supplementary Fig. 3a). Moreover, RT-qPCR demonstrated significantly higher expression of several viral lytic genes including ORF57, PAN, ORF52, and ORF50

**Fig. 2 Knockdown of UPF1 and UPF3X enhances KSHV lytic reactivation in iSLK.219 cells. a** Schematic experimental setup. iSLK.219 cells were transfected with indicated siRNAs for 48 h and then treated with or without Dox for 72 h. HEK293T cells were infected with supernatants of reactivated iSLK.219 cells. **b** Knockdown efficiency of UPF1 in iSLK.219 cells was determined by RT-qPCR. **c** GFP and RFP were imaged 48 h post-Dox treatment. Bar indicates 250 μm. **d** RFP-positive cells in **c** were quantified by flow cytometry. **e** RNA was extracted 24 h post-Dox treatment and expression of the indicated genes was quantified by RT-qPCR. **f** Cell lysates from latent and 48 h-Dox-induced cells were analyzed by immunoblotting for the indicated proteins (representative of three independent experiments). **g** Supernatant of iSLK.219 cells treated with indicated siRNAs were collected 72 h post-Dox treatment and virions were quantified by RT-qPCR. **h** Quantification of GFP-positive cells in HEK293T cells infected with supernatants derived from lytic iSLK.219 cells treated with the indicated siRNAs. **i** Quantification of LANA gene expression by RT-qPCR in HEK293T cells infected with supernatants derived from lytic iSLK.219 cells treated with the indicated siRNAs. **j** Knockdown efficiency of UPF3X in iSLK.219 cells was determined by RT-qPCR. **k** GFP and RFP were imaged 48 h post-Dox treatment. Bar indicates 250 μm. **l** RFP-positive cells in **k** were quantified by flow cytometry. **m** RNA was extracted 24 h post-Dox treatment and expression of the indicated genes was quantified by RT-qPCR. **n** Cell lysates from latent and 48 h-Dox-induced cells were analyzed by immunoblotting for the indicated proteins (representative of three independent experiments). **o** Supernatant of iSLK.219 cells treated with indicated siRNAs were collected 72 h post-dox addition and virions were quantified by qPCR. **p** Quantification of GFP-positive cells in HEK293T cells infected with supernatants derived from lytic iSLK.219 cells treated with the indicated siRNAs. **q** Quantification of LANA gene expression in HEK293T cells infected with supernatants derived from lytic iSLK.219 cells treated with the indicated siRNAs. Data are presented as mean values ± SD ($n = 3$ biologically independent samples). p-values were determined by the two-tailed Student's t-test. Source data are provided in Source Data file.

(RTA) in siUPF1-treated cells relative to siCON-treated cells (Fig. 2e). In addition, western blot analyses confirmed an increase in the protein levels of RTA, ORF59, and bZIP (Fig. 2f).

It is important to note that the primers that quantify RTA expression are specific to the viral-encoded RTA and do not detect the RTA-transgene within the cellular genome. Lytic reactivation in the iSLK.219 system is mediated by the Dox-induced expression of the chromosomally-encoded RTA thus we tested whether expression of the Dox-induced RTA transgene is affected by UPF1 knockdown. We treated uninfected iSLK cells with either siCON or siUPF1 and quantified expression of the RTA-transgene following Dox induction. Importantly, UPF1 depletion did not affect the expression of the Dox-inducible RTA transgene (Supplementary Fig. 4a, b), indicating the effect of NMD inhibition on KSHV lytic reactivation is independent of the transgene.

Next, we determined the effect of depleting UPF1 on virion production. siCON- or siUPF1-treated iSLK.219 cells were induced for 72 h whereupon culture supernatants were collected and viral genomes were quantified by qPCR. We observed a significant increase in supernatant virions from cells that had been treated with siUPF1 relative to siCON (Fig. 2g). Furthermore, leveraging a supernatant transfer assay coupled to flow cytometry we observed a 12-fold increase in infected GFP-positive cells when supernatants from UPF1-depleted cells were used to infect uninfected HEK293T cells (Fig. 2c, h and Supplementary Fig. 3b). Consistent with the increase in GFP-positive cells expression of the viral latent gene, latency associated nuclear antigen (LANA), was expressed significantly higher in HEK293T cells that were infected with supernatants from UPF1-depleted cells (Fig. 2i). Collectively, these results demonstrate that depletion of the core NMD factor UPF1 results in a significant enhancement in KSHV lytic reactivation and virion production.

Given that KSHV directly manipulates host RNA stability through expression of the viral-encoded RNA endonuclease SOX we sought to investigate the role of NMD in the context of SOX-mediated host shutoff[35]. To test this, we determined the effect of UPF1 knockdown in iSLK.219 cells harboring a well-characterized SOX-mutant (P176S) in which its RNase activity is significantly impaired[36,37]. Expression of P176S SOX in iSLK.219 cells results in an increased expression of multiple SOX target genes, including β-actin and eIF-1α, when compared to cells expressing a SOX allele with the mutation repaired (P176S MR, Supplementary Fig. 5a, b). UPF1 knockdown in cells expressing either P176S or the rescue mutant resulted in an enhancement of viral lytic gene expression relative to control-siRNA-treated cells. However, in P176S SOX-mutant cells the

enhancement of viral gene expression was reduced relative to the mutant rescue (Supplementary Fig. 5c–g).

**Knockdown of UPF3X enhances KSHV reactivation.** UPF1 plays a role in multiple RNA decay pathways. Thus, to further test the contribution of NMD to KSHV lytic reactivation we tested the effect of silencing UPF3X on viral gene expression and virion production in iSLK.219 cells. UPF3X is a component of the EJC and is involved in the activation of NMD; however, it is important to note that not all NMD substrates are UPF3X-dependent[6,13,38–40]. iSLK.219 cells were treated with Control- or UPF3X-specific siRNA (siUPF3X) and viral gene expression was quantified by RT-qPCR. While no increase in spontaneous reactivation was detected upon UPF3X depletion (Supplementary Fig. 6a–c), we did observe a significant increase in Dox-mediated KSHV reactivation. Quantification of lytic reactivation by flow cytometry indicated a 5-fold increase in lytic reactivation in siUPF3X-treated cells relative to siCON-treated cells (Fig. 2j–l and Supplementary Fig. 3c). Moreover, the expression of viral lytic genes ORF57, PAN, ORF52, and ORF50 was significantly higher in siUPF3X-treated cells relative to siCON-treated cells (Fig. 2m), and western blot analyses confirmed an increase in the protein levels of RTA, ORF59, and bZIP (Fig. 2n).

Next, we quantified virion production from siUPF3X- and siCON-treated iSLK.219 cells. Quantification of viral genomes in the culture supernatants by qPCR demonstrated a significant increase in virion production when cells are treated with siUPF3X (Fig. 2o). In addition, by supernatant transfer coupled with flow cytometry we observed a ~5-fold increase in infectious virions and this was further confirmed by RT-qPCR quantification of LANA expression in the infected cells (Fig. 2k, p, q and Supplementary Fig. 3d).

**NMD restricts KSHV lytic reactivation in PEL.** KSHV is the etiological agent of PEL, thus we sought to test the role of NMD in the KSHV-positive PEL cell line TREx-BCBL1-RTA. Consistent with our results from iSLK.219 cells depletion of UPF1 in latent TREx-BCBL1-RTA cells significantly increased the expression of several KSHV lytic mRNAs (Fig. 3a, b). PEL cells undergo a low level of spontaneous lytic reactivation at steady state thus it's possible that the observed increase in viral gene expression is within this population of cells. To test whether UPF1 depletion in TREx-BCBL1-RTA cells increases spontaneous reactivation we quantified the percentage of cells expressing the viral lytic noncoding RNA polyadenylated nuclear RNA (PAN) by RNA fluorescent in situ hybridization flow cytometry

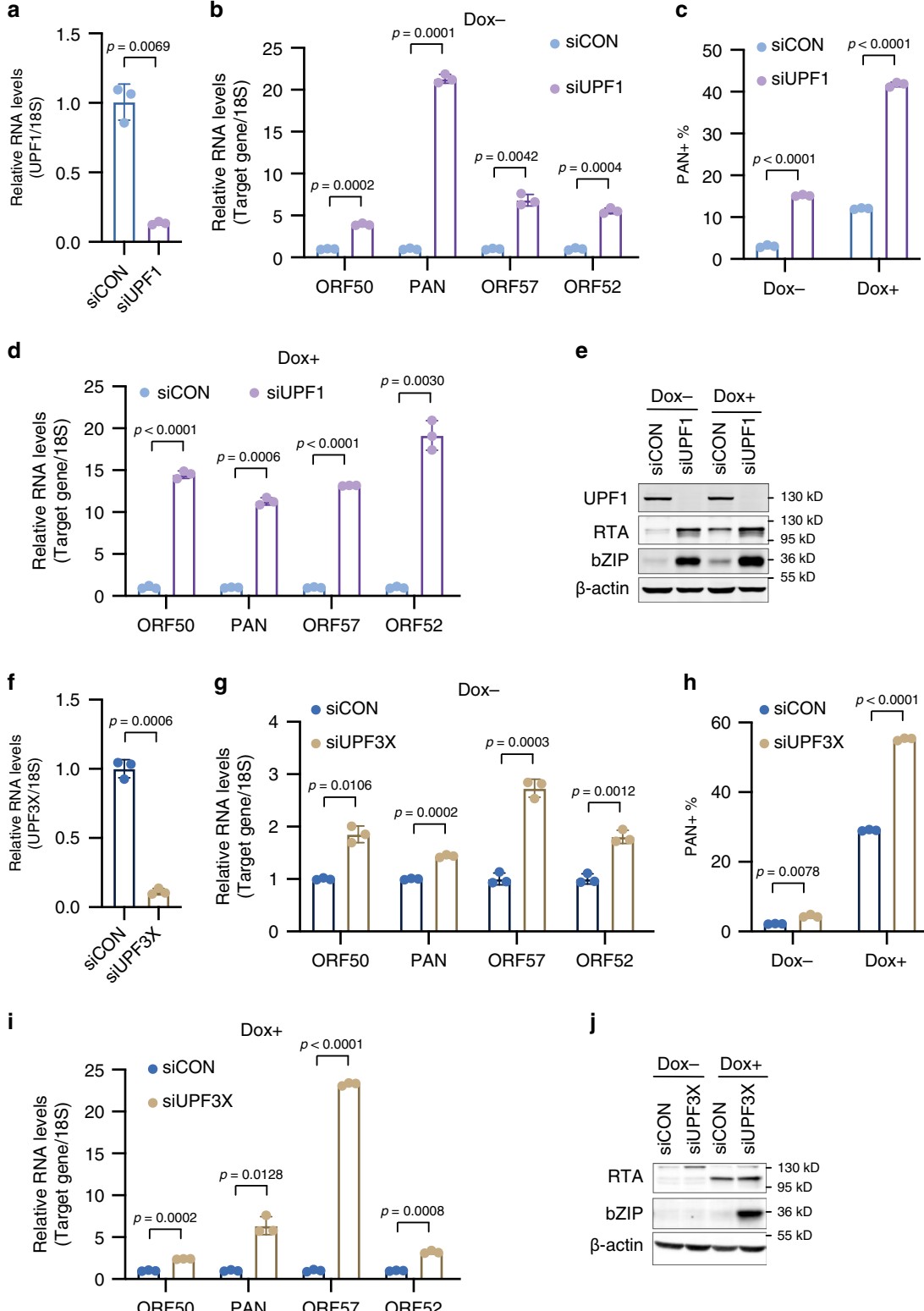

(RNA FISH-FLOW). While ~3% of siCON-treated TREx-BCBL1-RTA cells expressed PAN RNA, its expression was observed in ~14% of siUPF1-treated cells (Fig. 3c and Supplementary Fig. 7a). Thus, in both iSLK.219 and TREx-BCBL1-RTA cells knockdown of UPF1 increases spontaneous reactivation.

The effects of UPF1 knockdown on Dox-induced lytic gene expression in TREx-BCBL1-RTA cells was also determined. Total RNA and protein was isolated from siCON- or siUPF1-treated

cells at 24 and 48 h post-induction, respectively, and viral gene expression was quantified by RT-qPCR and western blot analyses (Fig. 3d, e). UPF1 depletion resulted in a significant increase in lytic RNA expression compared to siCON-treated cells, and an increase in the protein levels of RTA and bZIP were confirmed by western blot analyses. In addition, by RNA FISH-FLOW we observed a ~4-fold increase in PAN RNA expressing lytic cells 24 h post-induction (Fig. 3c).

**Fig. 3 Knockdown of UPF1 and UPF3X enhances lytic reactivation in TREx-BCBL1-RTA cells. a** Knockdown efficiency of UPF1 in TREx-BCBL1-RTA cells was determined by RT-qPCR. **b** Quantification of viral gene expression by RT-qPCR in latent TREx-BCBL1-RTA cells treated with the indicated siRNAs. **c** Quantification of PAN RNA expressing TREx-BCBL1-RTA cells by PAN FISH-FLOW in latent and 24 h post-Dox induction. **d** Quantification of viral gene expression by RT-qPCR in TREx-BCBL1-RTA cells treated with the indicated siRNAs 24 h post-dox induction. **e** Cell extracts from latent and 48 h-Dox-induced TREx-BCBL1-RTA cells treated with the indicated siRNAs were immunoblotted for viral proteins (representative of three independent experiments). **f** Knockdown efficiency of UPF3X in TREx-BCBL1-RTA cells was determined by RT-qPCR. **g** Quantification of viral gene expression by RT-qPCR in latent TREx-BCBL1-RTA cells treated with the indicated siRNAs. **h** Quantification of PAN RNA expressing TREx-BCBL1-RTA cells by PAN FISH-FLOW in latent and 24 h post-Dox induction. **i** Quantification of viral gene expression by RT-qPCR in TREx-BCBL1-RTA cells treated with the indicated siRNAs 24 h post-dox induction. **j** Cell extracts from latent and 48 h-Dox-induced TREx-BCBL1-RTA cells treated with the indicated siRNAs were immunoblotted for viral proteins (representative of three independent experiments). Data are presented as mean values ± SD ($n = 3$ biologically independent samples). $p$-values were determined by the two-tailed Student's $t$-test. Source data are provided in Source Data file.

---

We next sought to determine the effect of UPF3X depletion on KSHV reactivation in TREx-BCBL1-RTA cells. UPF3X depletion resulted in a minor increase in lytic viral gene expression as well as PAN RNA expressing cells, suggesting a minor increase in spontaneous reactivation (Fig. 3f–h). However, upon Dox induction we observed a robust increase in the expression of several lytic genes at both the RNA and protein level (Fig. 3i, j). Furthermore, we observed ~2-fold increase in PAN RNA expressing lytic cells by RNA FISH-FLOW (Fig. 3h and Supplementary Fig. 7b). Collectively, these data demonstrate that NMD restricts lytic gene expression in PEL and reveal a role for NMD in suppressing both spontaneous and Dox-induced lytic reactivation.

**Transcriptome-wide identification of NMD targets in PEL.** Phospho-UPF1 (p-UPF1) is a prominent marker of NMD substrates as its presence commits an RNA to degradation[17,18,41,42]. To identify NMD substrates in PEL cells we performed fRIP-coupled to high-throughput sequencing (fRIP-seq) using antibodies directed against p-UPF1 and, as a negative control, rabbit IgG (IgG). The p-UPF1 antibody has been extensively validated and prior to building sequencing libraries from the eluate we verified its ability to selectively enrich known NMD targets (Supplementary Fig. 8a)[41]. fRIP-seq was performed on extracts prepared from both latent and 48 h lytic TREx-BCBL1-RTA cells (Fig. 4a). We validated our fRIP-seq as previously described. For example, we confirmed enrichment of p-UPF1 to the 3′-UTRs of transcripts (Fig. 4d), and RNAs from 89 previously validated NMD targets were significantly enriched in libraries from p-UPF1-bound samples relative to all mRNAs (Fig. 4b, c)[8,41,42].

We identified 1863 and 2153 cellular transcripts enriched in p-UPF1 binding in latent and lytic cells, respectively (Supplementary Data 3 and 4). To determine whether NMD preferentially targets specific classes of transcripts in PEL we performed gene set enrichment analysis (GSEA) (Fig. 4e, f). We observed a significant enrichment in several GSEA hallmarks suggesting that NMD plays a widespread role in PEL cell biology. Interestingly, several of these hallmarks have been previously associated with the regulation of KSHV lytic reactivation[43–45].

While the majority of the p-UPF1 bound RNAs are of a coding RNA biotype annotation nearly one-third of the transcripts are annotated as noncoding RNA (Fig. 4g, h). This result is not entirely unexpected as several short and long noncoding RNAs, as well as pseudogenes, have previously been identified as NMD substrates[46–48]. Given that NMD is translation-dependent these results suggest the presence of noncoding RNA-derived translation products in PEL cells.

To determine whether viral RNAs are targeted for NMD we mapped fRIP-seq reads to KSHV annotated features (e.g., ORFs and repeats). We identified 24 annotated features preferentially enriched in p-UPF1 fRIP-seq data (Fig. 4i). The detection of some

lytic genes in the latent fRIP-seq data is likely due to the small percentage of PEL cells that are undergoing spontaneous reactivation. Interestingly, some of the features that are enriched in the latent samples are not present in the lytic samples. While the mechanisms behind this are unclear it is possible that spurious transcripts derived from latent genomes or spontaneous reactivation results in unique transcripts structures that are surveyed by NMD.

**ORF50 mRNA is associated with p-UPF1.** Intriguingly, we noticed that the several of the viral ORFs enriched by p-UPF1 are expressed from genomic regions with overlapping transcription units that terminate at a common polyadenylation site[49,50]. Additionally, the p-UPF1 reads are preferentially enriched in the most 3′ ORF (Supplementary Fig. 9a–d). Two prominent examples of this are the ORF50-K8-K8.1 and ORF45-48 loci where the p-UPF1 fRIP-seq reads are enriched over K8-K8.1 and ORF45-46, respectively (Fig. 5a, b). p-UPF1 preferentially binds the 3′ end of NMD substrates and thus we hypothesized that the longer, upstream transcripts, i.e., ORF50 and ORF48, are the NMD targets. In support of this, the ORF50 and ORF48 3′UTRs are exceptionally long as well as contain intronic sequences, both of which are prominent triggers of NMD. We note that the intron in the ORF48 3′UTR is derived from the intron we discovered in ORF46 (Fig. 1b, c and Supplementary Fig. 1a).

The protein encoded by the ORF50 transcript, RTA, is required for KSHV reactivation thus we focused our following experiments on determining the impact of NMD on its regulation and function. To test if ORF50 is associated with p-UPF1 in TREx-BCBL1-RTA cells we performed fRIP coupled to RT-qPCR using primers located near its transcription start site (TSS) (Fig. 5c). Indeed, we observed significant enrichment of ORF50 in p-UPF1 fRIP eluates. Importantly, the RNA polymerase III transcript, vault RNA 1-1, and viral genes ORF4 and ORF55, which were not identified as NMD targets by fRIP-seq, were not enriched by p-UPF1, while the positive control, cellular GADD45A, was. Additionally, using primers located near the TSS of ORF48, we also observed its enrichment in p-UPF1 eluates. These results demonstrate that both upstream initiating transcripts, i.e., ORF50 and ORF48, are bound by p-UPF1.

**NMD targets ORF50 mRNA via its 3′UTR.** We next sought to identify features within the ORF50 RNA that activates NMD. As noted above, the 3′UTR of ORF50 is exceptionally long and harbors 3′UTR introns and thus we tested whether its 3′UTR was sufficient to confer NMD sensitivity in a heterologous context. We inserted the 3′UTR of ORF50 downstream of a Renilla luciferase (RLuc) gene within a dual-luciferase construct and quantified expression of RLuc in transfected HEK293T cells (Fig. 5d). As we also observed p-UPF1 enrichment of the ORF48 transcript its 3′UTR was also tested. As a negative control, we inserted the 3′

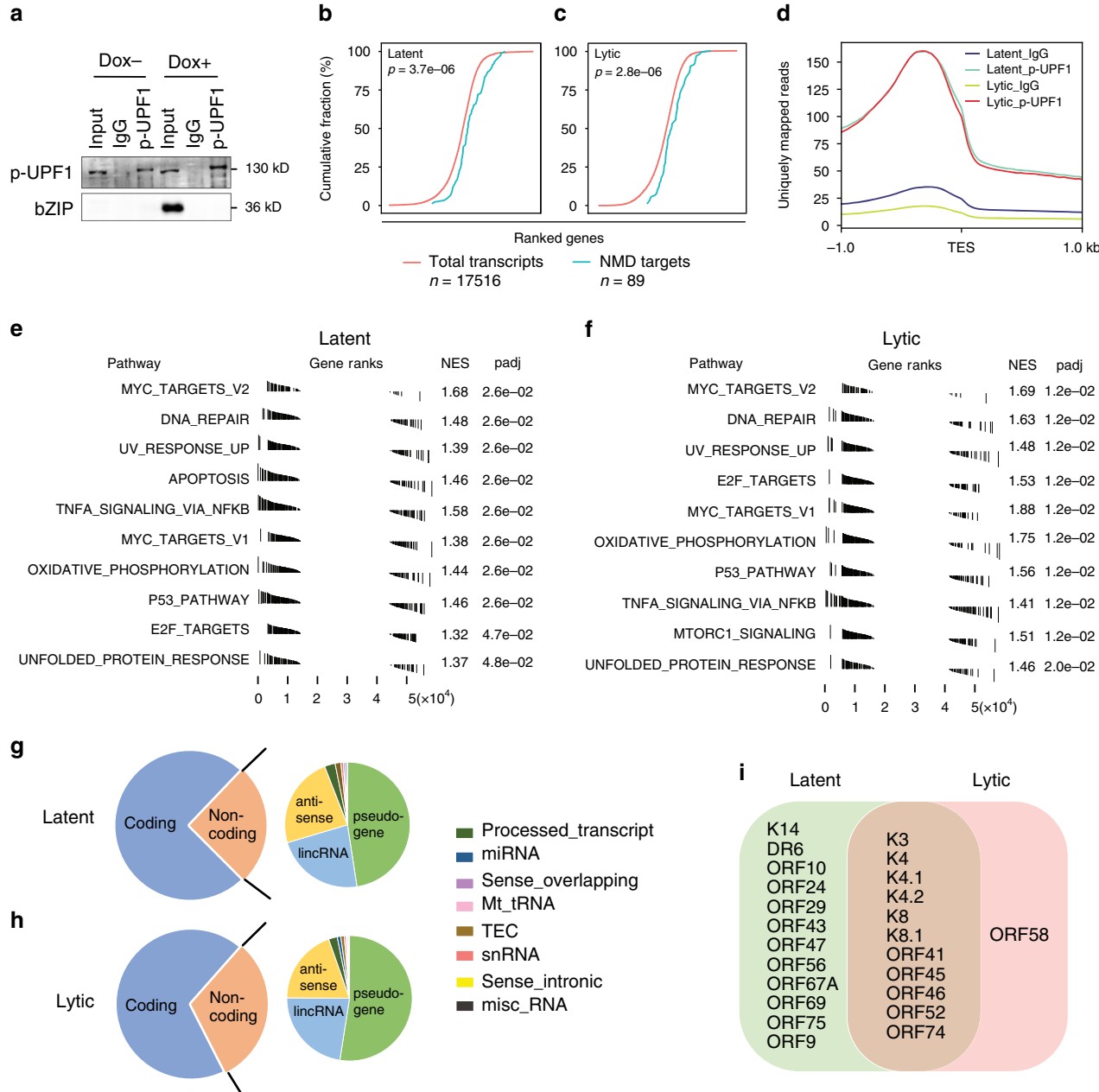

**Fig. 4 Identification of NMD targets in PEL by p-UPF1 fRIP-seq. a** Western blotting of p-UPF1 fRIP in latent (Dox−) and lytic (Dox+) TREx-BCBL1-RTA cells (representative of three independent experiments). **b, c** Cumulative fraction gene rank by fold enrichment of anti-p-UPF1 immunoprecipitation relative to anti-rabbit IgG immunoprecipitation in latent (**b**) and lytic (**c**) samples. *P*-value derived from the Wilcoxon rank-sum test. **d** Exonic paired-end sequence reads ±1 kilobase (kb) to either side of the transcript end site (TES) in 10-nucleotide bins in p-UPF1 and IgG immunoprecipitation. **e, f** Enriched GSEA hallmarks of latent (**e**) and lytic (**f**) p-UPF1 fRIP-seq datasets. **g, h** Pie chart representation of gene biotypes identified by latent (**g**) and lytic (**h**) p-UPF1 fRIP-seq. **i** Venn diagram of viral gene features enriched in latent and lytic p-UPF1 fRIP-seq.

UTR of ORF4 downstream of RLuc as we did not observe ORF4 enrichment in p-UPF1 fRIP-seq data. Insertion of the ORF50 or ORF48 3′UTR resulted in a significant reduction in RLuc expression, whereas insertion of the 3′UTR of ORF4 did not. In addition, expression of RLuc containing the vector control 3′UTR was also not affected. Consistent with a role for NMD in the destabilization of ORF50 and ORF48 levels, UPF1 depletion increased the expression of RLuc constructs harboring the 3′UTRs of ORF50 and ORF48 (Fig. 5d).

We next tested whether the ORF50 3′UTR is sufficient to recruit p-UPF1. To test this, we transfected RLuc reporter constructs harboring the vector control 3′UTR, ORF4 3′UTR, or ORF50 3′

UTR into HEK293T cells and determined whether the transcripts associated with p-UPF1 by fRIP RT-qPCR (Fig. 5e, f). We observed ~90-fold enrichment of RLuc transcripts possessing the ORF50 3′ UTR in p-UPF1 fRIPs. In contrast, RLuc transcripts containing the control or ORF4 3′UTR were not enriched. Importantly, the differential enrichment was not due to IP efficiency as p-UPF1 immunoprecipitated the cellular NMD target, GADD45A, efficiently regardless of which vector was transfected. These data demonstrate that the ORF50 3′UTR is sufficient to recruit p-UPF1.

To further investigate the regulation of ORF50 expression by NMD we quantified the stability of either the RLuc ORF50 3′UTR transcript or of the full-length ORF50 transcript expressed

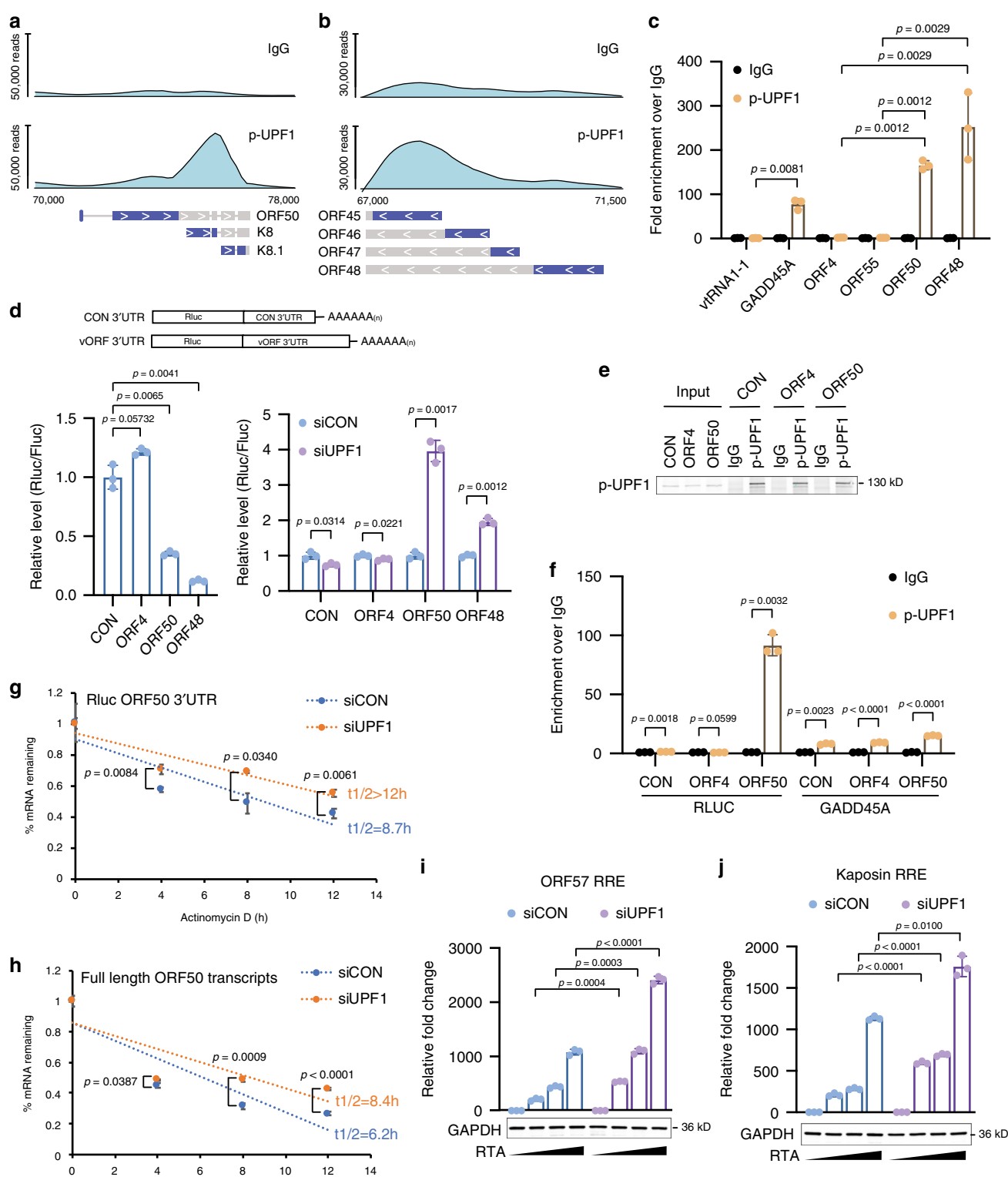

**Fig. 5 NMD targets ORF50 mRNA via its 3′UTR to repress RTA transactivation. a** Distribution of p-UPF1 fRIP-seq reads on loci of ORF50-K8.1 gene cluster. **b** Distribution of p-UPF1 fRIP-seq reads on ORF45-ORF48 gene clusters. **c** p-UPF1 fRIP-qPCR analysis of indicated viral genes in TREx-BCBL1-RTA cells 48 h post-Dox induction. **d** Luciferase assay of reporters with control or viral 3′UTR. **e** Western blot of p-UPF1 fRIP in cells transfected with luciferase reporter containing a control, ORF4 or ORF50 3′UTR. **f** p-UPF1 fRIP-qPCR analysis of luciferase reporter with control, ORF4 or ORF50 3′UTR. **g, h** Half-life measurement of the luciferase reporter mRNA with ORF50 3′UTR (**g**) and ORF50 mRNA (**h**) in control and UPF1-depleted HEK293T cells. **i, j** Luciferase assay quantifying RTA-mediated transactivation of ORF57 (**i**) and kaposin (**j**) in control and UPF1-depleted HEK293T cells. Data are presented as mean values ± SD ($n = 3$ biologically independent samples). $p$-values were determined by the two-tailed Student's $t$-test. Source data are provided in Source Data file.

heterologously in siCON or siUPF1-treated cells. Transfected cells were treated with actinomycin D (actD) to prevent new RNA synthesis and the level of remaining transcripts at 0, 4, 8, and 12 h after actD addition were quantified by RT-qPCR (Fig. 5g, h). We observed a significant delay in the degradation kinetics of RLuc ORF50 3′UTR transcript as well as the full-length ORF50 transcript in UPF1-depleted cells compared with siCON-treated cells. Thus, KSHV transcripts are targeted by NMD and the 3′ UTR of ORF50 is sufficient to confer NMD sensitivity in a heterologous context. These are the first DNA virus NMD-eliciting features to be identified.

Intron-containing and long 3′UTRs are known to trigger NMD. Thus, we investigated whether the introns within the ORF50 3′UTR are required to confer NMD susceptibility. To test this, a full-length ORF50 expression construct was transfected into HEK293T cells treated with siCon or siUPF1 and the levels of spliced and unspliced transcripts were quantified by RT-qPCR. Depletion of UPF1 resulted in an increase in the levels of ORF50 spliced mRNA whereas the unspliced transcripts remained unchanged (Supplementary Fig. 10a–f). Moreover, the ORF50 3′UTR introns were required to confer NMD sensitivity to RLuc reporter transcripts (Supplementary Fig. 10g). Although removal of the 3′UTR introns reduced its overall size, the ORF50 3′UTR is still exceptionally long at over 1.2 kb. Thus, these data strongly suggest that ORF50 3′UTR-mediated NMD is primarily intron dependent. Lastly, we investigated the ability of NMD-sensitive and NMD-immune ORF50 transcripts to reactive TREx-BCBL1-RTA cells. Consistent with observed role for the 3′UTR introns in mediating ORF50 decay, transfection of NMD-immune ORF50 into TREx-BCBL1-RTA cells was significantly more robust at facilitating KSHV lytic reactivation compared with NMD-sensitive ORF50 (Supplementary Fig. 10h).

Given that NMD regulates the expression of the ORF50 transcript we next tested whether inhibition of NMD was sufficient to potentiate RTA-mediated transcription. To test this, siCON- or siUPF1-treated HEK293T cells were cotransfected with the full-length ORF50 expression vector and luciferase reporter plasmids containing RTA-responsive elements (RRE) derived from the ORF57 and Kaposin genes. RTA-mediated transactivation of both promoters was significantly enhanced in siUPF1-treated cells (Fig. 5i, j). Collectively, these data demonstrate that the 3′UTR of ORF50 is sufficient to recruit UPF1 and activate NMD, thereby limiting RTA levels and reducing KSHV gene expression.

**NMD connects the cellular UPR pathway to ORF50 transcription**. Having determined NMD regulates ORF50 mRNA levels we were intrigued by the observation of an unfolded protein response (UPR) hallmark in our p-UPF1 GSEA as the UPR-activated transcription factor, X-box binding protein 1 (XBP1), has been demonstrated to transactivate the ORF50 promoter[51,52]. Thus, we hypothesized that NMD regulated the transcriptional activation of ORF50 via the UPR pathway in addition to its control of ORF50 mRNA stability. Spliced XBP1 (sXBP1), which is the active form of the transcription factor, is normally present at low levels. However, upon activation of the UPR the endoribonuclease inositol-requiring enzyme 1α (IRE1α) catalyzes the removal of an intron from XBP1 mRNA, resulting in a frameshift within the coding region leading to translation of the larger active sXBP1 isoform[53]. To test whether NMD influences expression of sXBP1 we evaluated its expression in siUPF1- and siCON-treated TREx-BCBL1-RTA cells. We observed an increase in both the mRNA and protein levels of sXBP1 in UPF1-depleted cells relative to siCON-treated cells (Fig. 6a, b). This effect was observed in both latent and Dox-induced lytic cells, and is likely caused by the moderate upregulation of IRE1α (Fig. 6c).

sXBP1 has been reported to bind the ORF50 promoter in PEL cells. Thus, we tested whether activation of the IRE1α-XBP1 branch in UPF1-depleted cells resulted in an increase in sXBP1 occupancy at the ORF50 promoter. By chromatin immunoprecipitation (ChIP) we observed an increase of sXBP1 on the ORF50 promoter in siUPF1-treated cells relative to control in both latent and lytic cells, implicating the UPR pathway in the spontaneous reactivation observed upon UPF1 depletion (Fig. 6d, e). Importantly, inhibition of IRE1α using the small molecule KIRA6, which prevents XBP1 splicing, reduced the presence of sXBP1 at the ORF50 promoter[54] (Fig. 6d-h). Consistent with the reduction in sXBP1 ORF50 promoter occupancy in KIRA6 treated cells, the expression of ORF50 mRNA was concomitantly reduced (Fig. 6i).

Next, we tested whether NMD-mediated activation of the IRE1α-XBP1 branch quantitatively effected the number of lytically reactivating cells by PAN RNA FISH-FLOW. Consistent with a role for NMD-mediated activation of sXBP1 in ORF50 expression and lytic reactivation, KIRA6 treatment resulted in a significant reduction in the levels of PAN RNA expressing cells (Fig. 6j and Supplementary Fig. 11). Collectively, these data demonstrate that NMD regulates activation of XBP1 splicing, and that loss of NMD triggers sXBP1 production and an increase in its recruitment to the ORF50 promoter facilitating lytic reactivation. Thus, NMD regulates both transcription and post-transcriptional stability of ORF50 mRNA to restrict KSHV reactivation (Fig. 6k).

## Discussion

NMD is an evolutionarily conserved RNA decay mechanism that has recently garnered attention as a cell-intrinsic restriction mechanism of RNA viruses. However, whether NMD plays a role during DNA virus infection has remained unknown. Here, we have determined that NMD is a potent restriction mechanism against the DNA virus KSHV. We find that siRNA-mediated silencing of the NMD factors UPF1 and UPF3X significantly enhances KSHV lytic reactivation in both iSLK.219 and TREx-BCBL1-RTA cells. Leveraging p-UPF1 fRIP-seq we identified NMD targets transcriptome-wide in both latent and lytic PEL cells and targets include both host and viral RNAs. Remarkably, the mRNA encoding RTA, the master transcription factor governing KSHV reactivation, is targeted by NMD via its 3′UTR and silencing of UPF1 is sufficient to potentiate RTA-mediated transactivation. Moreover, we find that NMD suppresses activation of the UPR and production of the transcription factor sXBP1, which can transactivate the ORF50 promoter in PEL. Thus, our results have uncovered a mechanism whereby NMD regulates RTA expression at both the transcriptional and post-transcriptional level. Beyond KSHV, our study broadens the spectrum of viruses that NMD restricts and highlights the biological importance of NMD as a universal antiviral defense mechanism against both DNA and RNA viruses.

We were emboldened to investigate the role of NMD in the KSHV lifecycle in part by our observation of a highly complex pre-mRNA splicing landscape within the KSHV transcriptome. While only 27 introns were previously described, our study identified 372 and 68 introns within iSLK.219 and TREx-BCBL1-RTA cells, respectively[31]. These analyses coupled with recent efforts to map the TSS landscape of KSHV underscores that we still lack a complete annotation of the KSHV transcriptome, and that viral transcript architecture is cell-type specific[28].

KSHV encodes many overlappingly genes with shared usage of a downstream polyadenylation signal, and ~45% and ~73% of these gene clusters harbor introns in TREx-BCBL1-RTA and iSLK.219 cells, respectively. Many of the novel introns identified

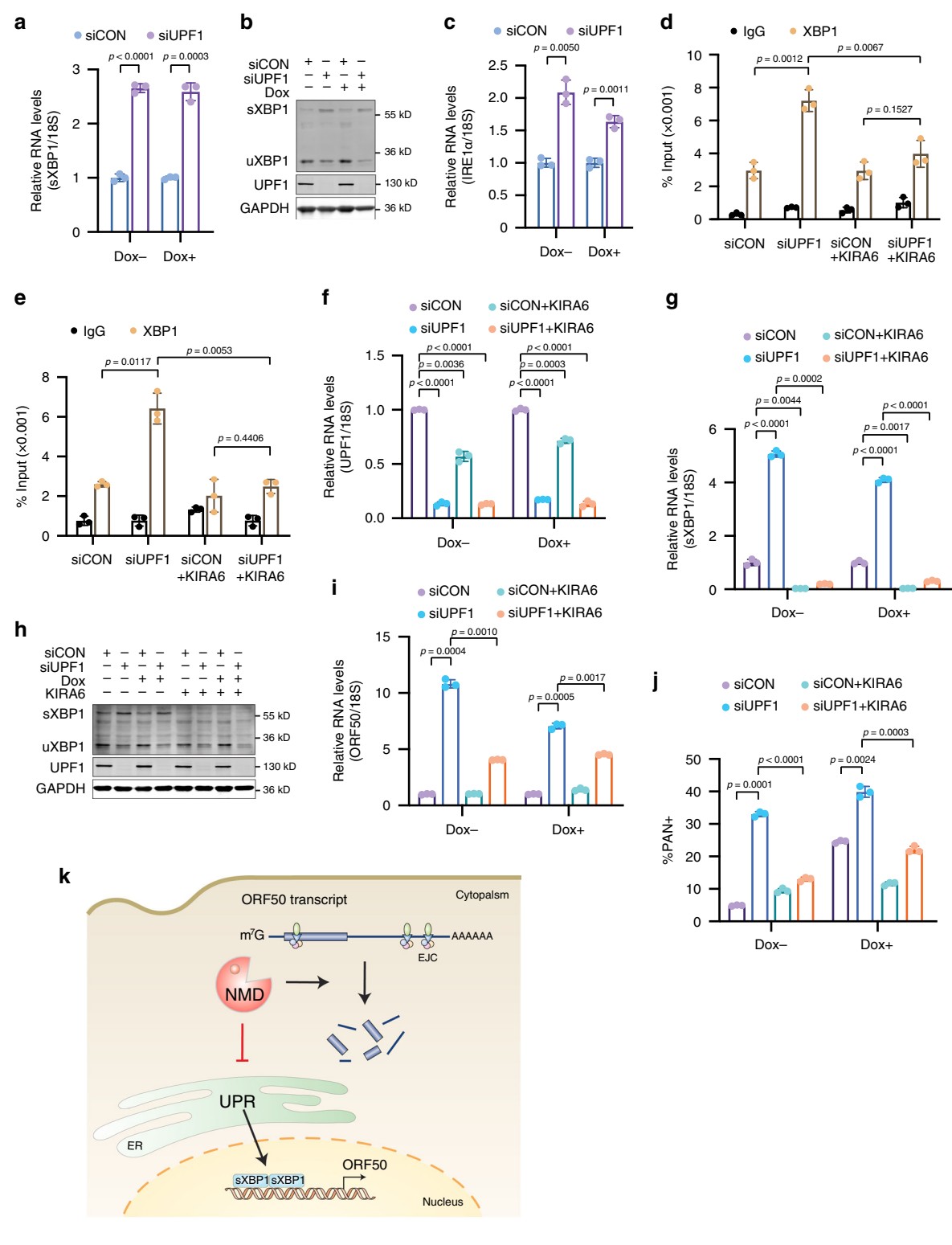

are present within protein coding ORFs as well as in the 3′UTR of the 5′ proximal gene in polycistronic RNAs. For example, the intron we validated in the ORF46 coding sequence is simultaneously positioned within the 3′UTR of ORF48. The presence of an intron with an RNA affords additional opportunities for the cell to regulate the RNAs expression, such as through the activation of NMD when located within a 3′UTR. Along this line, our experiments demonstrate that the ORF48 3′UTR, which harbors the ORF46 intron, is sufficient to confer NMD sensitivity in a heterologous context. Moreover, >90% of the viral RNAs enriched

**Fig. 6 NMD targets the cellular UPR pathway to restrict transcription of ORF50. a** Quantification of sXBP1 expression by RT-qPCR in latent (Dox−) and lytic (Dox+) TREx-BCBL1-RTA cells treated with the indicated siRNAs. **b** Western blot of cell lysates prepared from latent (Dox−) and lytic (Dox+) TREx-BCBL1-RTA cells. sXBP1, spliced XBP1; uXBP1, unspliced XBP1. **c** Quantification of IRE1α expression by RT-qPCR in latent (Dox−) and lytic (Dox+) TREx-BCBL1-RTA cells treated with the indicated siRNAs. **d, e** Quantification of XBP1 occupancy at the ORF50 promoter in latent (**d**) and lytic (**e**) TREx-BCBL1-RTA cells by ChIP-qPCR. Signals were normalized to input. **f, g** RNA was isolated from the indicated TREx-BCBL1-RTA cells and the expression of UPF1 (**f**) and sXBP1 (**g**) was quantified by RT-qPCR. **h** Western blot of cell lysates prepared from of latent (Dox−) and lytic (Dox+) TREx-BCBL1-RTA cells in the condition of UPF1 depletion and KIRA6 treatment. **i** Quantification of ORF50 expression in the indicated TREx-BCBL1-RTA cells. **j** Quantification of PAN RNA expressing cells by PAN FISH-FLOW in latent (Dox−) and lytic (Dox+) TREx-BCBL1-RTA cells following UPF1 depletion and KIRA6 treatment. **k** The model depicting how NMD regulates the expression of ORF50 at both transcriptional and post-transcriptional level. Data are presented as mean values ± SD ($n = 3$ biologically independent samples). $p$-values were determined by the two-tailed Student's $t$-test. Source data are provided in Source Data file.

by p-UPF1 are expressed from genomic loci that harbor polycistronic transcripts and thus resemble mRNAs containing PTCs. Interestingly, however, there are many viral polycistronic transcripts that were not enriched by p-UPF1 fRIP, suggesting there exist mechanisms for viral transcript-specific evasion of NMD. The identification of these mechanisms is likely to shed light on both KSHV-host interactions as well as mechanisms of NMD-evasion.

NMD regulates the expression of the ORF50 mRNA, which encodes the master regulator RTA, at both the transcriptional and post-transcriptional level. The post-transcriptional regulation of ORF50 is mediated through its 3′UTR, which harbors several features known to stimulate NMD, such as a 3′UTR > 1 kb as well as 3′UTR introns[49,50,55]. Along this line, the ORF50 3′UTR is sufficient to promote p-UPF1 association and confer NMD sensitivity. Interestingly, we identified the UPR as a GSEA hallmark in p-UPF1 fRIP-seq data and this pathway has been previously linked to the transcriptional activation of the ORF50 promoter[51,56,57]. Indeed, we find that inhibition of NMD resulted in the production of the UPR transcription factor sXBP1 and its occupancy at the ORF50 promoter. Furthermore, inhibition of IRE1α, the endonuclease required for XBP1 splicing, prevented the increase of sXBP1 occupancy at the ORF50 promoter that was observed upon UPF1 silencing. Thus, the transcriptome-wide profiling of NMD targets in PEL facilitated the identification of a novel mechanism connecting the UPR to ORF50 transcription and KSHV reactivation.

sXBP1 plays a fundamental role in B-cell differentiation into antibody-secreting plasma cells[53,58]. The presence of a functional sXBP1 transcription factor binding site within the ORF50 promoter couples KSHV reactivation from latency with a B cells attempt to differentiate into an antibody-secreting cell. Interestingly, PEL cells appear phenotypically similar to B cells arrested immediately prior to differentiation into plasma cells[59,60]. Whether there is a role for NMD in specifying this intermediate cell state of PEL is interesting and warrants further investigation.

In addition to the UPR pathway, other hallmarks were enriched that have been previously associated with KSHV reactivation. For example, MYC and mTOR GSEA hallmarks are observed in the p-UPF1 fRIP-seq data and these factors have been linked to the KSHV lifecycle[43,44,61,62]. We anticipate that future investigations into how NMD regulates these pathways will uncover additional crosstalk between KSHV reactivation and NMD.

## Methods

**Cells and viruses.** iSLK.219[29] (kindly provided by Dr. Britt Glaunsinger, University of California, Berkeley) and HEK293T (ATCC) were grown in Dulbecco's modified Eagle medium (DMEM; Invitrogen) supplemented with 10% fetal bovine serum (FBS; Invitrogen). TREx-BCBL1-RTA cells[30] (kindly provided by Dr. Britt Glaunsinger, University of California, Berkeley) were maintained in RPMI 1640 medium (Invitrogen) supplemented with 10% FBS (Invitrogen) and 2 mM L-glutamine (Invitrogen). All cells were maintained with 100 U of penicillin/ml and 100

μg of streptomycin/ml (Invitrogen) at 37 °C under 5% $CO_2$. iSLK.219 and TREx-BCBL1-RTA cells were lytic reactivated with 1 μg/ml of doxycycline (Dox; Fisher Scientific). The TREx-BCBL1-RTA cells were treated with 5 μM KIRA6 (Cayman chemical) to inhibit the activity of IRE1α.

**siRNA knockdowns.** iSLK.219 cells were transfected at 60–80% confluency with 40 nM UPF1- or UPF3X- siRNA (sequences in Supplementary Table 1) or MISSION siRNA Universal Negative Control #1 (Sigma) using Lipofectamine RNAi-Max (Invitrogen). Forty-eight hour post-transfection cells were reactivated as described above.

TREx-BCBL1-RTA cells were microporated using Neon transfection system (Invitrogen) with 100 pmol indicated siRNA twice at a 48 h interval at 1600 v, 10 ms pulse width, and three pulses. Twenty-four hour post microporation, cells were reactivated as described above. KIRA6 was added at the final concentration of 5 μM after the second microporation.

**Viral infection.** For supernatant transfer, iSLK.219 cells were reactivated with Dox for 72 h, after which the supernatant was collected and supplemented with 8 μg/ml polybrene. HEK293T cells were spinfected with the supernatant at $1000 \times g$ for 1 h at room temperature (RT). The infection media was replaced with fresh media and incubated for 48 h, followed by flow-cytometry analysis.

**Flow cytometry.** iSLK.219 transfected with indicated siRNA and HEK293T cells infected with KSHV were fixed with 2% paraformaldehyde and then analyzed on a BD LSR Fortessa or Canto II instrument. Data were analyzed with FlowJo X software (TreeStar). The gating strategy is shown in Supplementary Fig. 12a, b.

**Fluorescence in situ hybridization and flow cytometry.** TREx-BCBL1-RTA cells were fixed in 4% (vol/vol) paraformaldehyde for 30 min at RT, washed with PBS-FISH buffer (1X PBS, 0.2 mg/ml RNase-free BSA) twice, and then permeabilized with 1X PBS containing 0.2% (vol/vol) Tween-20 for another 30 min at RT. The permeabilized cells were then hybridized with Alexa-Fluor 488 or Alexa-Fluor 647 labeled PAN anti-sense oligos (sequences in Supplementary Table 1) in HB 10% dx buffer (10% (wt/vol) dextran sulfate, 2× saline-sodium citrate (SSC), 10% (vol/vol) formamide, 1 mg/ml tRNA and 0.2 mg/ml BSA) at 37 °C overnight. After extensive washing with HBW buffer (2× SSC, 10% (vol/vol) formamide and 0.2 mg/ml RNase-free BSA), cells were analyzed on BD Canto II instrument. Data were analyzed with FlowJo X software (TreeStar). The gating strategy is shown in Supplementary Fig. 12c.

**RT-qPCR.** Total RNA was isolated with TRIzol (Invitrogen) in accordance with the manufacturer's instructions. RNA was DNase I (NEB) treated at 37 °C for 20 min and inactivated with EDTA at 70 °C for 10 min. cDNA was synthesized from DNase-treated RNA with random 9-mer (Integrated DNA Technologies) and M-MLV Reverse Transcriptase (Promega). qPCR was performed using the PowerUp SYBR Green qPCR kit (Thermo Scientific) with appropriate primers (Supplementary Table 1).

**Western blotting.** Whole-cell lysates were prepared with lysis buffer (50 mM Tris [pH 7.6], 150 mM NaCl, 0.5% NP-40) and quantified by Bradford assay (BioRad). Equivalent amounts of each sample were resolved by SDS-PAGE, electro-transferred to PVDF membrane (Millipore), and blotted for the indicated proteins. Antibodies: UPF1 (Abcam, #ab109363, 1:10000), p-UPF1 (Ser1127, EMD Millipore, #07-1016, 1:1000), eIF4A3 (Bethyl, A302-981A, 1:5000), GAPDH (Invitrogen, GA1R, #MA5-15738, diluted 1:5000), β-actin (Invitrogen, BA3R, #MA5-15739, 1:1000), ORF50 and ORF59 (1:10,000, kindly provided by Dr. Britt Glaunsinger, University of California, Berkeley), bZIP (1:2000, kindly provided by Dr. Cyprian Rossetto, University of Nevada, Reno), FLAG (Thermo Fisher Scientific, FG4R, MA1-91878, 1:1000), XBP1 (Abcam, ab220783, 1:1000). Primary antibodies were followed by Alexa-Fluor 680-conjugated secondary antibodies

(Life Technologies, goat anti-rabbit #A27042, goat anti-mouse #A28183, 1:10,000) and visualized by Odyssey CLx imaging system (LI-COR).

**Luciferase assays**. HEK293T cells were transfected with psicheck2 and its derivative constructs using PolyJet in vitro DNA transfection reagent (Signagene laboratories). Twenty-four hour post-transfection cells were collected and used to measure both renilla and firefly luciferase activity using dual-luciferase reporter assay system (Promega) on a GLOMAX 20/20 Luminometer (Promega).

**RNA half-life determination**. HEK293T cells at 60–80% confluency were transfected with the indicated siRNA using Lipofectamine RNAiMax (Invitrogen). Twenty-four hour later cells were transfected with the indicated constructs using PolyJet in vitro DNA transfection reagent (Signagene laboratories). Twenty-four hour post-plasmid transfection, Actinomycin D (at a final concentration of 7.5 μg/ml) was added to the cells and RNA was collected at 0, 4, 8, and 12 h post-addition. RT-qPCR were performed as described above.

**Chromatin immunoprecipitation (ChIP)**. Ten million TREx-BCBL1-RTA cells with indicated treatment were cross-linked with 1% formaldehyde for 10 min and then quenched by 0.125 M glycine for 5 min. Fixed cells were collected and lysed in SDS lysis buffer (1% SDS, 10 mM EDTA, 50 mM Tris, pH 8.1). Chromatin DNA was fragmented to ~200–1000 bp in length with the Covaris LE220 sonicator (Covaris). After removing the insoluble material by $12,000 \times g$ spinning at 4 °C, one-tenth of the sonicated lysates were diluted with 9 volume of dilution buffer (0.01% SDS, 1.2 mM EDTA, 16.7 mM Tris, pH 8.1, 1.1% Triton-X100, 167 mM NaCl) and incubated overnight with 3 μg of anti-XBP1 antibody or rabbit IgG per reaction and then with Protein G Dynabeads (BioRad) for 1 h. After extensively washing, beads were incubated at 65 °C overnight to reverse formaldehyde cross-link. Following protease K and RNase A treatment, immunoprecipitated DNA was purified using a PCR Purification kit (QIAGEN) and analyzed by qPCR. All qPCR signals were normalized to the input. The sequences of the PCR primers used for amplification are listed in Supplementary Table 1.

**fRIP-seq**. Latent and 48 h post-Dox-induced TREx-BCBL1-RTA cells were crosslinked with 0.1% formaldehyde for 10 min in PBS, and unreacted formaldehyde was neutralized with 0.3 M glycine for 5 min. Cells were washed 2X with PBS, then resuspended in RIPA buffer (50 mM Tris, pH 8.0, 1% IGEPAL CA 630, 0.5% sodium deoxycholate, 0.05% SDS, 1 mM EDTA, 150 mM NaCl, 1 mM DTT, and RNase and protease inhibitors) and kept on ice for 10 min. Cell lysates were briefly sonicated and centrifuged to generate soluble cell extracts, which were then incubated with IgG or p-UPF1 (Ser1127, EMD Millipore, #07-1016) antibody at 4 °C overnight. Protein G magnetic beads were added and incubated at 4 °C for another 2 h. Beads were then washed three times for 10 min at 4 °C in RIPA buffer containing 0.1% SDS, 1 M NaCl, and 1 M urea. Protein–RNA crosslinks were reversed by adding 100 mM Tris, pH 8.0, 10 mM EDTA, 1% SDS, and 2 mM DTT to eluted samples and heating to 70 °C for 45 min. RNA was recovered by extraction with TRIzol and then again with phenol:chloroform:isoamyl alcohol [25:24:1 (vol/vol)] followed by ethanol precipitation. Paired-end RNA-sequencing libraries were prepared from the recovered RNA using the NEBNext Ultra II Directional RNA Library Prep Kit (NEB) according to the manufacture recommendations. Libraries were then subjected to paired-end sequencing on a Nova-Seq with 150 cycles at the Vanderbilt Technologies for Advanced Genomics (VANTAGE).

**RNA-seq and fRIP-seq data analysis**. Raw reads quality in fastq files were assessed by FastQC (https://www.bioinformatics.babraham.ac.uk/projects/fastqc/). Reads were aligned to the human reference genome (gencode GRCh38.p10) and KSHV genome (GQ994935.1) using STAR under 2-pass mode[63]. We estimated transcript abundance using Rsubread, and enriched transcripts were called using edgeR[64,65]. All the transcripts were summarized to biotypes annotated within GENCODE database (gencode.v24) with R scripts. Genome coverages were presented with Gviz or IGV viewer[66,67]. GSEA analysis was performed with R package fgsea (https://bioconductor.org/packages/release/bioc/html/fgsea.html) using hallmark genesets. For evaluation of fRIP read coverage at 3′ ends, uniquely mapped reads were extracted and transformed to bigwig file using bamCoverage command (deeptools) with the bin size of 10[68]. Then all the transcripts from gencode.v24 annotation were used for coverage analysis using computeMatrix (deeptools: https://deeptools.readthedocs.io/en/develop/content/list_of_tools.html) under reference-point mode (TES) with parameter—metagene set to TRUE. The enrichment of NMD target in UPF1 fRIP-seq experiment was tested with Wilcoxon rank-sum test. kpLogo was used to analyze nucleotide usage at the KSHV 5′ and 3′ splicing junctions[69]. Statistical analyses were done with R scripts.

**Reporting summary**. Further information on research design is available in the Nature Research Reporting Summary linked to this article.

## Data availability

Raw data are provided in the Source Data file. Sequencing data from this study have been deposited in SRA under project number PRJNA598976. High-throughput sequencing reads were aligned to the human reference genome (gencode GRCh38.p10, https://www.ncbi.nlm.nih.gov/assembly/GCF_000001405.36/) and KSHV genome (GQ994935.1). All the transcripts were summarized to biotypes annotated with GENCODE database (gencode.v24, ftp://ftp.ebi.ac.uk/pub/databases/gencode/Gencode_human/release_24/gencode.v24.chr_patch_hapl_scaff.annotation.gtf.gz). Custom code are available at github. A reporting summary for this article is available as a Supplementary file. Source data are provided with this paper.

## Code availability

All codes used in this study are available at github. Source data are provided with this paper.

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

## Acknowledgements

We would like to thank members of the Karijolich lab for insightful discussion. We thank Britt Glaunsinger (University of California, Berkeley,) Gary Hayward (John Hopkins University), Zhi-Ming Zheng (National Cancer Institute), and Cyprian Rossetto (University of Nevada, Reno) for providing antibodies against KSHV proteins. The Karijolich laboratory was supported by startup funds from Vanderbilt University Medical Center and National Institutes of Health (NIH) grants R01AI141448 (to J.K.), R01CA250051 (to

J.K.), and an American Cancer Society Research Scholar Award RSG MPC - 133907 (to J.K.). J.K. is a Pew Biomedical Scholar.

## Author contributions

Conceived and designed the experiments: J.K. and Y.Z. Performed the experiments: Y.Z., M.S., W.D., and Z.X. Analyzed the data: J.K., Y.Z., M.S., and X.Y. Wrote the manuscript: J.K. and Y.Z.

## Competing interests

The authors declare no competing interests.
