## [Peer Review File · Nature Communications]

Reviewers' comments:

Reviewer #1 (Remarks to the Author):

The major claims of this well-written manuscript are 1. previous splicing events have been missed by previous reports, 2. repression of NMD genes enhances lytic reactivation and virion production, 3. some important viral transcripts are targets of NMD. These claims will be interesting to the expanding NMD field and virologists. This report also brings to light another example of interactions with NMD factors and noncoding RNAs. The work is convincing with genomics-based sequencing methods followed by validation assays that included the important controls. It is important to distinguish between effect lytic reactivation and abortive reactivation by showing a new round of infection with virus from the NMD repression assays. They showed this striking result in Figure 2. These are some of the highest amounts of lytic reactivation that this reviewer has witnessed. A significant contribution in the many splicing events that have been mapped in this report. This challenges an incorrect assumption that many viral genes are not spliced. They have validated some of the new introns by cloning and sequencing the splice junctions. The statistical methods and details of the methods were fine.

Below are some specific comments that would improve this strong and thought-provoking manuscript.

1. What is the significance of NMD in the lytic cycle in the context of host shutoff?
2. Representative flow cytometry histogram plots should be included to complement the column graphs of RFP+ or PAN+ cells.
3. The GEO accession number appears to be from a previous manuscript. It is unknown if the current sequencing data have been deposited.
4. How can the authors determine whether repression of UPF1 and UPF3X is affecting lytic reactivation by NMD-independent mechanisms?
5. How was the antibody immunoprecipitation of p-UPF1 validated before fRIP-seq?
6. Lines 241-242: Have these “noncoding” RNAs been found to be associated with ribosomes in other reports?
7. Figure 5F: Measuring additional negative control regions would be helpful in Fig. 5 to show viral messages that were not found in fRIP-seq have low levels of detection by qPCR.

8. The spliced form of XBP1 is activated in some developmental stages of B cells, which are relevant to KSHV infections. This could be explored in the Discussion section.

Reviewer #2 (Remarks to the Author):

In this paper, Zhao et al. explore the effects of nonsense-mediated RNA decay on KSHV lytic reactivation. The paper makes three important claims. First, and most importantly, they claim that NMD “poses a significant barrier” to KSHV lytic reactivation. Second, they use transcriptomics to identify host and viral NMD targets in infected cells and to identify novel introns in viral genes. Third, they link NMD with the unfolded protein response and expression of the KSHV transcription factor RTA. The strength of the paper is the exciting and novel observation that NMD restricts viral lytic reactivation. The work supporting that conclusion is strong and rigorous. The weakness in the paper (discussed more below) is that the paper is full of potentially important hints at what the role and/or mechanisms of NMD in viral lytic reactivation, none of which are fully fleshed out to provide a solid conclusion. The paper would be significantly improved if they would focus on one potential driving factor (e.g. RTA as NMD substrate) and investigate it thoroughly. As it stands now, the paper conflates multiple loosely connected observations that do not support any single firm conclusion about any of the role(s) of NMD in KSHV reactivation.

Major concerns:

1. The role of NMD in KSHV lytic reactivation is the main idea of the paper. However, the paper starts with an in depth splicing analysis that is never tied into the rest of the paper in any concrete fashion. They imply that the 3' UTR splice sites may be driving NMD on viral transcripts but no substantiation of this is found anywhere in the paper. Even with RTA, which had previously described 3' UTR introns, no validation that splicing was linked to NMD was presented. For example, If they knockdown UPF1, do they see accumulation of spliced 3' UTR isoforms? Does removal of the 3' UTR intron abolish NMD, or is it driven by long 3' UTRs? Given its central role in KSHV lytic reactivation, it would be compelling to know whether expression of an NMD-insensitive RTA mRNA complements most of the NMD phenotypes. Currently, Figure 1 is a descriptive dataset only tangentially related to the rest of the paper.

2. Similarly, the host targets (Figure 4) and XBP1 data (Figure 6) do not lead to any solid conclusions. There's nothing wrong with the data in Figure 6, but they do not convincingly demonstrate that NMD regulation of XBP1 has any effect on the virus. The data say that XBP1 might be an important

NMD target related to KSHV reactivation. The data also show that upon lytic reactivation, there is a less than 2-fold difference in ORF50 RNA levels in the siUPF1 treatment plus/minus KIRA6. While this may indeed be biologically relevant, it's difficult to say if/how this change contributes to the robust effects of NMD on KSHV reactivation. Perhaps there is a link, but Figure 6 data seems more appropriate for a "Figure 1" of a new paper rather than a mechanistic conclusion (and model) for the current paper.

3. The fRIP studies are interpreted too strongly. In many cases, this is the only experimental support that the authors use to classify something as a "NMD target". It is well established that formaldehyde RIP approaches often has high background, so fRIP should be interpreted with considerable caution. Conversely, they are also seeing false negatives because it appears that approximately one-third of the 89 "NMD Targets" have greater signal in the IgG than the anti-UPF1p pulldowns (i.e. $\log_2FC < 0$) (Fig 4b/c). Moreover, rigorous validation needs to be done. For example, how many of these genes are up-regulated upon NMD inhibition? Admittedly, this is difficult (or even impossible) to do with viral lytic RNAs because of the confounding factor of increased reactivation upon NMD inhibition. However, this should not be a confounding issue with host RNAs or viral latent genes. The analysis is also confusing. Most CLIP-type experiments employ peak-calling algorithms, but it doesn't seem to be the case here. By eye, it appears that they find nearly 50% of their transcripts have more signal in anti-UPF1 than IgG (Fig 4b/4c) and they suggest thousands of genes are NMD targets. What are their cutoffs and how are they justified? This seems like significantly more than expected, but no comparisons to similar experiments by other groups are discussed so it's difficult to assess. Moreover, there's no correction for RNA levels. For example, for the ORF45 traces show what they interpret to be a 3' UTR peak (Fig 5). However, this could simply arise due to high background because the ORF45 transcript is expressed at higher levels than ORF46-48.

Other:

1. The fRIP-PCR assays require a control of a viral nonspliced (Fig 1) or non-NMD target mRNA in addition to vtRNA1-1.
2. In Figure 1, they "preference for a G nucleotide in the 3' of the splice acceptor site. The identification of consensus splice site sequences..." The 3' splice sites shown do not show a strong preference for AG. This is therefore clearly not representative of a consensus splice site.
3. More browser shots in supplemental of KSHV genes would help readers assess the fRIP peaks, particularly if compared to RNA-seq from the same time point.

Minor:

1. The GSE number is incorrect.
2. In Figure 1d, ORF45, the slash is at wrong place.
3. In figures such as 3b and 3d which are so similar, it could be helpful to give headers or color code according to if they are BCBL vs iSLK2.19

4. Figure 2m is UPF3X, called UPF1 in text.

Reviewer #3 (Remarks to the Author):

The impact of NMD on DNA virus infections is largely unexplored. In this study, the authors use global analyses to greatly expand (~10X) the number of splice junctions in the KSHV transcriptome, many of which were found in the 3' UTR and result in the deposition of the EJC complex as inferred by RIP experiments using eIF4A3 – hence leading them to examine a role for NMD. UPF1 KD resulted in a minor increase in lytic viral growth but interestingly a major increase in latent reactivation. The authors go onto to examine NMD targets globally and focus on the ORF50 transcript as a regulated example. Finally, the authors identify the UPR-associated ORF50 transcription factor sXBP1as another NMD-regulated factor during infection.

Overall the manuscript is well-written, the data are extremely convincing, and conclusions are well-supported. The study does a great job at establishing NMD as a factor that influences the regulation of a DNA virus infection and will be of very high interest and impact in the field.

I have only one very minor suggestion to polish the manuscript: The EJC label in the model figure (6K) is compacted/compressed and should be redrawn to enhance readability.

Reviewer #1

The major claims of this well-written manuscript are 1. previous splicing events have been missed by previous reports, 2. repression of NMD genes enhances lytic reactivation and virion production, 3. some important viral transcripts are targets of NMD. These claims will be interesting to the expanding NMD field and virologists. This report also brings to light another example of interactions with NMD factors and noncoding RNAs. The work is convincing with genomics-based sequencing methods followed by validation assays that included the important controls. It is important to distinguish between effect lytic reactivation and abortive reactivation by showing a new round of infection with virus from the NMD repression assays. They showed this striking result in Figure 2. These are some of the highest amounts of lytic reactivation that this reviewer has witnessed. A significant contribution in the many splicing events that have been mapped in this report. This challenges an incorrect assumption that many viral genes are not spliced. They have validated some of the new introns by cloning and sequencing the splice junctions. The statistical methods and details of the methods were fine.

Below are some specific comments that would improve this strong and thought-provoking manuscript.

1. **Comment:** What is the significance of NMD in the lytic cycle in the context of host shutoff?

Response: In this revised manuscript we now include supplementary data (Supplementary Figure 5) investigating the role of NMD in the context of WT and host shutoff defective KSHV lytic reactivation. We find that NMD inhibition in both conditions results in an enhancement of viral gene expression relative to control siRNA treated cells. However, in the absence of host shutoff the enhancement of viral gene expression is reduced relative to the WT virus.

2. **Comment:** Representative flow cytometry histogram plots should be included to complement the column graphs of RFP+ or PAN+ cells.

Response: We now include representative flow cytometry histogram plots for all column graphs of RFP+ GFP+ or PAN+ cells. The data is included in Supplementary Figure 2d, 3, 7 and 11.

3. **Comment:** The GEO accession number appears to be from a previous manuscript. It is unknown if the current sequencing data have been deposited.

Response: Thank you for pointing this out. The data is submitted to SRA with project number PRJNA598976. The data is set to be available about publication of the manuscript.

4. **Comment:** How can the authors determine whether repression of UPF1 and UPF3X is affecting lytic reactivation by NMD-independent mechanisms?

Response: Thank you for this comment. In fact, we are currently investigating the role of Staufen-mediated decay (SMD), which is a pre-mRNA splicing-independent RNA decay pathway that UPF1 functions in. However, these analyses are premature as well as out of the scope of this manuscript.

5. **Comment:** How was the antibody immunoprecipitation of p-UPF1 validated before fRIP-seq?

Response: The initial extensive validations of the antibody were conducted in the laboratory of Dr. Lynne Maquat, a pioneer in NMD (PMID: 25184677). Prior to our preparation of p-UPF1 fRIP-seq libraries we quantified the enrichment of two well characterized NMD targets and two control mRNAs in the p-UPF1 fRIP eluates by RT-qPCR. These RT-qPCRs are now shown in Supplementary Figure 8.

6. **Comment:** Lines 241-242: Have these “noncoding” RNAs been found to be associated with ribosomes in other reports?

Response: We are currently investigating the role of noncoding RNAs that are bound by p-UPF1 in TREx-BCBL1-RTA cells. However, this work is out of the scope of this manuscript and thus we do not include a further analysis of this.

7. **Comment:** Figure 5F: Measuring additional negative control regions would be helpful in Fig. 5 to show viral messages that were not found in fRIP-seq have low levels of detection by qPCR.

Response: We have now added several additional negative (non-NMD target) controls to our RT-qPCR analysis. Specifically, we tested the association of p-UPF1 to ORFs 4 and 55, which are non-NMD targets and do not possess introns, in Fig 5c. We did not observe association of p-UPF1 with either of these transcripts. In addition, we tested whether fusing the 3'-UTR of ORF4 to the luciferase reporter was sufficient to recruit p-UPF1 in Fig 5e. The ORF4 3'UTR was not capable of recruiting p-UPF1.

8. **Comment:** The spliced form of XBP1 is activated in some developmental stages of B cells, which are relevant to KSHV infections. This could be explored in the Discussion section.

Response: Thank you for this suggestion. We have now expanded our discussion to include additional information on XBP1 in B cell biology that is relevant to KSHV.

Reviewer #2

In this paper, Zhao et al. explore the effects of nonsense-mediated RNA decay on KSHV lytic reactivation. The paper makes three important claims. First, and most importantly, they claim that NMD “poses a significant barrier” to KSHV lytic reactivation. Second, they use

transcriptomics to identify host and viral NMD targets in infected cells and to identify novel introns in viral genes. Third, they link NMD with the unfolded protein response and expression of the KSHV transcription factor RTA. The strength of the paper is the exciting and novel observation that NMD restricts viral lytic reactivation. The work supporting that conclusion is strong and rigorous. The weakness in the paper (discussed more below) is that the paper is full of potentially important hints at what the role and/or mechanisms of NMD in viral lytic reactivation, none of which are fully fleshed out to provide a solid conclusion. The paper would be significantly improved if they would focus on one potential driving factor (e.g. RTA as NMD substrate) and investigate it thoroughly. As it stands now, the paper conflates multiple loosely connected observations that do not support any single firm conclusion about any of the role(s) of NMD in KSHV reactivation.

Major concerns:

- 1. Comment:** The role of NMD in KSHV lytic reactivation is the main idea of the paper. However, the paper starts with an in depth splicing analysis that is never tied into the rest of the paper in any concrete fashion. They imply that the 3' UTR splice sites may be driving NMD on viral transcripts but no substantiation of this is found anywhere in the paper. Even with RTA, which had previously described 3' UTR introns, no validation that splicing was linked to NMD was presented. For example, If they knockdown UPF1, do they see accumulation of spliced 3' UTR isoforms? Does removal of the 3' UTR intron abolish NMD, or is it driven by long 3' UTRs? Given its central role in KSHV lytic reactivation, it would be compelling to know whether expression of an NMD-insensitive RTA mRNA complements most of the NMD phenotypes. Currently, Figure 1 is a descriptive dataset only tangentially related to the rest of the paper.

Response: Thank you for the comments and in light of these we have added several additional analyses

- A. We have now added an analysis of intron location in Fig1d and Supplementary Figure 1b.
- B. We have performed the requested experiment to knockdown UPF1 and investigate accumulation of ORF50 spliced 3' UTR isoforms in Supplementary Figure 10a-f. We observe that knockdown of UPF1 results in accumulation of ORF50 spliced 3' UTR isoforms whereas the unspliced pre-mRNA isoforms are not affected.
- C. We have also investigated whether removal of the 3'UTR introns abolishes NMD in Supplementary Figure 10g. We observe that removal of the 3'UTR introns abolished the sensitivity of the transcript to UPF1 knockdown, indicating the splicing in 3'UTR is sufficient to confer NMD susceptibility.
- D. We have also investigated whether an NMD-insensitive RTA mRNA is more robust at facilitating KSHV lytic reactivation compared to a NMD-sensitive mRNA. This analysis is shown in Supplementary Figure 10h. We observe that lytic viral gene expression is enhanced when an ORF50 transcript lacking 3' introns (NMD-insensitive) is expressed compared to ORF50 transcript 3'UTR UTRs (NMD-sensitive).

2. **Comment:** There's nothing wrong with the data in Figure 6, but they do not convincingly demonstrate that NMD regulation of XBP1 has any effect on the virus. The data say that XBP1 might be an important NMD target related to KSHV reactivation. The data also show that upon lytic reactivation, there is a less than 2-fold difference in ORF50 RNA levels in the siUPF1 treatment plus/minus KIRA6. While this may indeed be biologically relevant, it's difficult to say if/how this change contributes to the robust effects of NMD on KSHV reactivation. Perhaps there is a link, but Figure 6 data seems more appropriate for a "Figure 1" of a new paper rather than a mechanistic conclusion (and model) for the current paper.

Response: The robust effects of NMD on reactivation are not strictly a result of XBP1. It is the combination of both the transcriptional and post-transcriptional effects of NMD on ORF50 expression. Figure 6 demonstrates that the inhibition of NMD leads to an increase in ORF50 expression via the activation of sXBP1. KIRA6 treatment, which targets the IRE1 α endonuclease, affects the production of sXBP1 and thus the transcriptional regulation of ORF50, not its post-transcriptional regulation. The biological effect of KIRA6 treatment is observed in our quantification of lytic reactivated TREx-BCBL1-RTA cells in Fig. 6j.

3. **Comment:** The fRIP studies are interpreted too strongly. In many cases, this is the only experimental support that the authors use to classify something as a "NMD target". It is well established that formaldehyde RIP approaches often has high background, so fRIP should be interpreted with considerable caution. Conversely, they are also seeing false negatives because it appears that approximately one-third of the 89 "NMD Targets" have greater signal in the IgG than the anti-UPF1p pulldowns (i.e. $\log_2FC < 0$) (Fig 4b/c). Moreover, rigorous validation needs to be done. For example, how many of these genes are up-regulated upon NMD inhibition? Admittedly, this is difficult (or even impossible) to do with viral lytic RNAs because of the confounding factor of increased reactivation upon NMD inhibition. However, this should not be a confounding issue with host RNAs or viral latent genes. The analysis is also confusing. Most CLIP-type experiments employ peak-calling algorithms, but it doesn't seem to be the case here. By eye, it appears that they find nearly 50% of their transcripts have more signal in anti-UPF1 than IgG (Fig 4b/4c) and they suggest thousands of genes are NMD targets. What are their cutoffs and how are they justified? This seems like significantly more than expected, but no comparisons to similar experiments by other groups are discussed so it's difficult to assess. Moreover, there's no correction for RNA levels. For example, for the ORF45 traces show what they interpret to be a 3' UTR peak (Fig 5). However, this could simply arise due to high background because the ORF45 transcript is expressed at higher levels than ORF46-48.

Response: We have broken down our responses below to make our response more clear.

We reject the assumption that it is "well established" that formaldehyde approaches have high background. fRIP-seq is a widely used method for profiling RNA-protein interactions and has revealed fundamental insight into many RNA-mediated gene regulatory processes (e.g. PMID: 26883116, PMID: 23317505, PMID: 30174290,

among many others). While high concentrations and prolonged treatment with formaldehyde can surely increase noise in the system, the conditions we employ in our experiments are similar to many others and are in fact well below the normal concentrations of formaldehyde used in Chromatin immunoprecipitation analyses. Moreover, our washes are extremely stringent and include RIPA buffer containing 0.1% SDS, 1 M NaCl, and 1 M urea.

There is a misinterpretation of cumulative distribution analyses (Fig 4b/c). The X-axis is the log₂ fold change ranking of transcripts enriched in p-UPF1 fRIP and not a true log₂ fold change. We have now changed the X-axis so that this more clear. Additionally, cumulative distribution analyses starts with an equal distribution assumption, thus it is incorrect to interpret these analyses by drawing a line up from 0 and estimating how many transcripts fall on either side---as appears to have been made based on the comment "*it appears that they find nearly 50% of their transcripts have more signal in anti-UPF1 than IgG*". What is important is the ranking of transcripts and position of the distributions with regard to each other. As is clearly observed in Fig 4b/c, the distribution of the known NMD targets is shifted towards the right (more positive numbers). The significance of this shift is tested by Mann-Whitney test ($p = 3.7e-6$ and $2.8e-6$ for latent and lytic, respectively). Similar analyses of p-UPF1 bound RNAs can be found in (PMID25184677 Fig 1B-E and PMID30275517 Fig 1c).

Comment: The analysis is also confusing. Most CLIP-type experiments employ peak-calling algorithms, but it doesn't seem to be the case here.

Response: Peak calling is not normally performed on fRIP-seq analyses. Moreover, an RNase digestion step was not performed prior to fRIP-seq and therefore peak calling would not be informative.

Comment: Moreover, rigorous validation needs to be done. For example, how many of these genes are up-regulated upon NMD inhibition? Admittedly, this is difficult (or even impossible) to do with viral lytic RNAs because of the confounding factor of increased reactivation upon NMD inhibition. However, this should not be a confounding issue with host RNAs or viral latent genes. What are their cutoffs and how are they justified?

This seems like significantly more than expected, but no comparisons to similar experiments by other groups are discussed so it's difficult assess. Moreover, there's no correction for RNA levels. For example, for the ORF45 traces show what they interpret to be a 3' UTR peak (Fig 5). However, this could simply arise due to high background because the ORF45 transcript is expressed at higher levels than ORF46-48.

Response: Transcripts are considered as a NMD target if they met two criteria: 1) It was ≥ 2 -fold ($\log_2 FC \geq 1$) enriched in p-UPF1 fRIP over the IgG fRIP, and 2) The adjusted p-value is less than 0.05 ($p_{adj} < 0.05$). The genes shown in Supplemental Table 3 and 4 all meet these criteria.

We observe 17516 cellular genes expressed in latent TREx-BCBL1-RTA cells and classify 1863 as NMD targets (as shown in Fig 4b). This equates to ~10% of the cellular transcriptome as NMD targets and is in line with other studies that have attempted to define the NMD regulome and have placed this number at 5-10% of transcriptome. Considering cell-type specific transcriptomes and various experimental and bioinformatic analyses our data support the conclusion that roughly 5-10% of the transcriptome is surveyed by NMD.

When analyzing RIP-seq data there is the choice of whether to normalize to IgG or Input. We have normalized to IgG through our manuscript as this is 1) a common way of analyzing RIP-seq data, and 2) All fRIP RT-qPCR analyses throughout the manuscript are normalized to IgG. However, to clearly demonstrate that our analyses are accurate we have now added the input (RNA extraction from cell lysis of fRIP experiment) track of fRIP for several viral genes including ORF45-ORF48. p-UPF1 is clearly enriched at the 3' end of genes even compared with Input. Moreover, quantification of the p-UPF1 fRIP-seq reads over the ORF45-48 locus again identifies ORF45 and ORF46 as NMD targets.

Other:

- 4. Comment:** The fRIP-PCR assays require a control of a viral nonspliced or non-NMD target mRNA in addition to vtRNA1-1.

Response: We have now added several additional negative (non-NMD target) controls to our RT-qPCR analysis. Specifically, we tested the association of p-UPF1 to ORFs 4 and 55, which are non-NMD targets and do not possess introns, in Fig 5c. We did not observe association of p-UPF1 with either of these transcripts. In addition, we tested whether fusing the 3'-UTR of ORF4 to the luciferase reporter was sufficient to recruit p-UPF1 in Fig 5e,f. The ORF4 3'UTR was not capable of recruiting p-UPF1.

- 5. Comment:** In Figure 1, they “preference for a G nucleotide in the 3' of the splice acceptor site. The identification of consensus splice site sequences...” The 3' splice sites shown do not show a strong preference for AG. This is therefore clearly not representative of a consensus splice site.

Response: We have now reworded our text to include additional text and citations regarding cellular splice-site conservation as well as our interpretation of the kpLogo analysis. Specifically, on line 114 we state: “*Analyses of eukaryotic pre-mRNA splicing have revealed the 5' and 3' splice-sites predominantly consist of GU and AG dinucleotide sequences. However, deviations from this are well known to exist.*”

Additionally, on lines 119, we have reworded our text to state: “*The identification of a consensus 5' splice site as well as sequences that can serve as 3' splice acceptors flanking KSHV introns suggests these events are facilitated by the cellular spliceosome.*”

6. Comment: More browser shots in supplemental of KSHV genes would help readers assess the fRIP peaks, particularly if compared to RNA-seq from the same time point.

Response: We have now added several new IGV browser shots. These are included in the Supplementary Figure 9.

7. Comment: The GSE number is incorrect.

Response: Thank you for pointing this out. The data is submitted to SRA and the project number is PRJNA598976. The data is set to be available about publication of the manuscript.

8. Comment: In Figure 1d, ORF45, the slash is at wrong place.

Response: This is now corrected.

9. Comment: In figures such as 3b and 3d which are so similar, it could be helpful to give headers or color code according to if they are BCBL vs iSLK2.19

Response: The role of UPF1 and UPF3X in iSLK.219 and TREx-BCBL1-RTA cells are separated into two different figures. In Fig2 we investigate the role of these proteins in iSLK.219 cells. In Fig3 we investigate the role of these proteins in TREx-BCBL1-RTA cells. Fig3b and 3d are RT-qPCR analyses of latent and lytic TREx-BCBL1-RTA cells, respectively. To make it more readable, they are now labeled as “Dox-” and “Dox+”.

10. Comment: Figure 2m is UPF3X, called UPF1 in text.

Response: Thank you for pointing this out. We have now corrected this in the text.

Reviewer #3:

The impact of NMD on DNA virus infections is largely unexplored. In this study, the authors use global analyses to greatly expand (~10X) the number of splice junctions in the KSHV transcriptome, many of which were found in the 3' UTR and result in the deposition of the EJC complex as inferred by RIP experiments using eIF4A3 – hence leading them to examine a role for NMD. UPF1 KD resulted in a minor increase in lytic viral growth but interestingly a major increase in latent reactivation. The authors go onto to examine NMD targets globally and focus on the ORF50 transcript as a regulated example. Finally, the authors identify the UPR-associated ORF50 transcription factor sXBP1 as another NMD-regulated factor during infection. Overall the manuscript is well-written, the data are extremely convincing, and conclusions are well-supported. The study does a great job at establishing NMD as a factor that influences the regulation of a DNA virus infection and will be of very high interest and impact in the field.

I have only one very minor suggestion to polish the manuscript: The EJC label in the model

figure (6K) is compacted/compressed and should be redrawn to enhance readability.

1. **Comment:** The EJC label in the model figure (6K) is compacted/compressed and should be redrawn to enhance readability.

Response: Thank you for pointing this out. We have now redrawn this to enhance readability.

REVIEWERS' COMMENTS:

Reviewer #1 (Remarks to the Author):

Overall, the authors have addressed my concerns. They have included additional results, controls, and analysis. My enthusiasm for this manuscript has improved with these additions and it is clear that this work will lead to future interesting studies.

1. The authors now include new information in Sup. Fig. 5 and lines 175-186 that addresses my previous comment.
2. The additional plots are useful additions.
3. The GEO accession number has been updated, but was not available to reviewers.
4. It seems worth mentioning that UPF1 plays a role in multiple decay pathways. This could be done without giving away current plans about Staufen-mediated decay.
5. The validation data now included in Sup. Fig. 8 is useful.
6. The authors claim that stating whether certain non-coding RNAs have been reported in other ribosome-associated reports is beyond the scope of this work. It appears to this reviewer that context of their finding would be helpful to readers.
7. The authors include additional negative controls in their assays, which strengthens their previous conclusion.
8. The authors have now added relevant information about the context of XBP1 in B cell development to the Discussion Section.

Reviewer #2 (Remarks to the Author):

This is good work with broad interest to the virology field. The authors have addressed all of my concerns. I found two typos:

1. "rending" line 85

2. "preformed" line 290.

REVIEWERS' COMMENTS:

Reviewer #1 (Remarks to the Author):

Overall, the authors have addressed my concerns. They have included additional results, controls, and analysis. My enthusiasm for this manuscript has improved with these additions and it is clear that this work will lead to future interesting studies.

Comment 1. The authors now include new information in Sup. Fig. 5 and lines 175-186 that addresses my previous comment.

Response: Thank you.

Comment 2. The additional plots are useful additions.

Response: Thank you.

Comment 3. The GEO accession number has been updated, but was not available to reviewers.

Response: Accession numbers is now available.

Comment 4. It seems worth mentioning that UPF1 plays a role in multiple decay pathways. This could be done without giving away current plans about Staufen-mediated decay.

Response: Thank you. On line 207 we now include the sentence: "UPF1 plays a role in multiple decay pathways."

Comment 5. The validation data now included in Sup. Fig. 8 is useful.

Response: Thank you.

Comment 6. The authors claim that stating whether certain non-coding RNAs have been reported in other ribosome-associated reports is beyond the scope of this work. It appears to this reviewer that context of their finding would be helpful to readers.

Response: Thank you for this comment. However, this would require a significant amount of work for us to analyze several other ribosome-profiling data sets (all in different cell types than the one used here) and intersect that data with p-UPF1 fRIP-seq data. We do not see the usefulness in such an enormous amount of work especially considering it is out of scope. Our description of our data is more general and just reflects that noncoding RNAs have been found on ribosomes in the

Comment 7. The authors include additional negative controls in their assays, which strengthens their previous conclusion.

Response: Thank you.

Comment 8. The authors have now added relevant information about the context of XBP1 in B cell development to the Discussion Section.

Response: Thank you.

Reviewer #2 (Remarks to the Author):

This is good work with broad interest to the virology field. The authors have addressed all of my concerns. I found two typos:

1. "rending" line 85
2. "preformed" line 290.

Response: Thank you. We have corrected both typos in the text.